# IS IT NECESSARY TO INJECT CAUSALITY INTO CHAIN-OF-THOUGHT REASONING?

## ABSTRACT

The integration of Chain-of-Thought into large language models has advanced their reasoning capabilities. However, how CoT produces correct answers through stepwise reasoning—and why it often makes mistakes—remains poorly understood, as the causality between reasoning steps is often difficult to quantify. This limitation raises the open question: *Is it necessary to inject causality into CoT reasoning?* In this paper, we formalize the CoT as a structural causal model, representing the reasoning process as a causal graph to complete the mathematical modeling. On this basis, we develop a step-level causal correction algorithm, Causalizing Chain-of-Thought (CauCoT), which identifies causally erroneous steps in CoT (i.e., incorrect or unintelligible steps) based on the defined CoT Average Causal Effect, and iteratively updates them until all steps are causally correct—a state we define as relaxed causal correctness. Given the lack of datasets for evaluating the impact of causality on CoT reasoning, we release the Causal Reasoning Benchmark (CRBench), the first benchmark targeting causal errors in CoT, which comprises both causally labeled real CoT reasoning error and newly generated CoT with injected causal errors. Experimental results on LLMs demonstrate that CauCoT can efficiently correct causal errors in CoT and improve the understandability of reasoning. We inject causality into CoT reasoning from mathematical, algorithmic, dataset-driven, and empirical levels, thereby providing strong evidence for the necessity of causality in achieving correct and interpretable stepwise reasoning.

## 1 INTRODUCTION

> "We do not have knowledge of a thing until we grasped its cause."
>
> — Aristotle

Large language models (LLMs) Liu et al. (2021); Yang et al. (2024a); Guo et al. (2025) have emerged as cornerstones of modern AI systems, revolutionizing problem-solving through their emergent ability to perform stepwise reasoning, known as Chain-of-Thought (CoT). CoT bridges raw computational power and structured problem-solving by decomposing complex tasks into stepwise reasoning traces designed to maintain correctness Kojima et al. (2022); Hu et al. (2023); Chen et al. (2024); Sprague et al. (2024); Yeo et al. (2025a). While CoT has driven substantial advancements in reasoning tasks, it often fails to generate human-understandable reasoning Yeo et al. (2025a) and frequently produces erroneous steps, thereby limiting both accuracy and interpretability Lanham et al. (2023); Sprague et al. (2024). This raises a critical need to uncover the mechanism of CoT in order to improve its correctness and interpretability. Recent studies have attempted to uncover the mechanisms behind CoT reasoning, primarily through empirical and statistical analyses of factors such as the upper bounds of reasoning capability Feng et al. (2024); Chen et al. (2024), the generalizability of reasoning Li et al. (2024b); Yao et al. (2025); Yang et al. (2025), or the effect of reasoning length on performance Li et al. (2024); Chen et al. (2025); Yeo et al. (2025b). However, these studies offer limited insight into the fundamental mechanisms through which CoT produces correct and coherent reasoning.

To move beyond observational characterizations of CoT, we turn to causality—a foundational paradigm for understanding and improving decision-making systems Pearl (2009); Yao et al. (2021). Causality has already proven useful in enhancing the trustworthiness and generalization of machine learning models in domains including robust prediction Li et al. (2024a); Xie et al. (2024a), multimodal reasoning Wang et al. (2024); Tai et al. (2024), and agent-based planning Abdulaal et al.

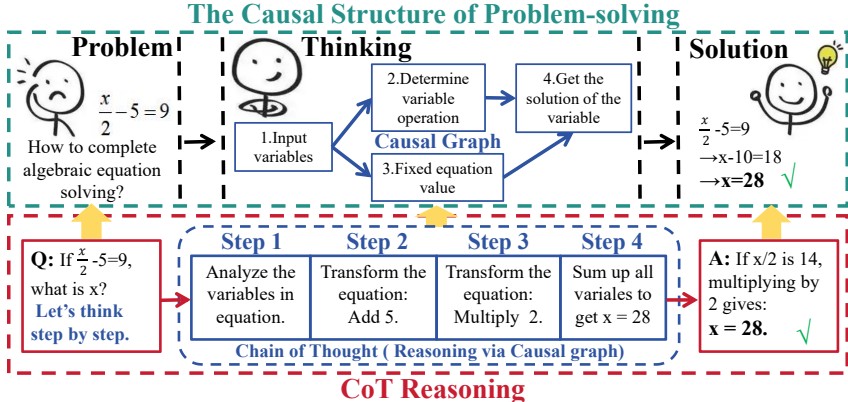

Figure 1: We hypothesize that CoT reasoning aligns with the causality of problem-solving. For instance, algebraic reasoning depends on identifying structural causal relations between variables, i.e., the causal graph. Similarly, CoT mirrors these causal relations, leading to the correct answer.

(2024); Li et al. (2025b). Naturally, applying causality to enhance CoT reasoning is emerging as a promising direction Kıcıman et al. (2023); Jin et al. (2023b); Bhattacharjee et al. (2024), with early efforts targeting tasks such as knowledge-based reasoning Wu et al. (2024) and causal question answering Jin et al. (2023a); Zhang et al. (2024a). Yet, emerging evidence suggests that LLMs often imitate causal patterns without genuine causal understanding Zečević et al. (2023); Wu et al. (2024); Babu Shrestha et al. (2025), raising concerns about whether CoT reasoning can genuinely benefit from causal injection. This gap leads us to an open question:

> *Is it necessary to inject causality into CoT reasoning?*

This question arises from the fact that causal dependencies between reasoning steps are often implicit and difficult to quantify. To address this, we identify two key gaps in current research:

1. The lack of models to identify causal relations for CoT steps limits the understandability of reasoning.
2. The lack of algorithms to implement step-level causal correction for CoT undermines the correctness and interpretability of reasoning.

In this paper, we begin with **mathematical modeling**, assuming that CoT mirrors the causality Rubin (1980); Pearl (2009); Kaddour et al. (2022) of problem-solving (as illustrated in Figure 1). We formalize the stepwise structure of CoT as a Structural Causal Model (SCM). This formulation represents each reasoning step as a node in a directed causal graph and injects causality into the modeling of CoT. Building on this formulation, we proceed to **algorithmic design** with Causalizing Chain-of-Thought (CauCoT)—a step-level causal correction algorithm. We define the CoT Average Causal Effect (CACE) to quantify the causal influence of each reasoning step from two complementary perspectives: the contributory evidence and the logical continuity. This metric enables CauCoT to identify causally erroneous steps—those that are either logically incorrect or unintelligible—and iteratively refine the reasoning trace until all steps are both interpretable and correct, reaching a state we define as relaxed causal correctness. By leveraging CACE to identify and correct causally erroneous steps, CauCoT injects causality into the steps of CoT reasoning. To address the absence of causal annotations for step-level errors in existing CoT datasets, we undertake **benchmark construction** by publishing the first benchmark for causal errors in CoT—Causal Reasoning Benchmark (CRBench). Based on four defined common types of causal errors in CoT, we construct CRBench by causally labeling existing CoT process-error benchmarks and generating new high-quality CoT reasoning data with injected causal errors. In doing so, we inject causality into CoT reasoning at the dataset-driven level, enabling evaluation of causal correctness. Finally, through extensive **empirical evaluation** on multiple open-source LLMs, we demonstrate that causally informed reasoning significantly improves both correctness and interpretability, thus injecting causality into the empirical CoT reasoning. These four components—mathematical modeling, algorithmic design, benchmark construction, and empirical evaluation—progressively inject causality into CoT at the *mathematical*, *algorithmic*, *dataset-driven*,

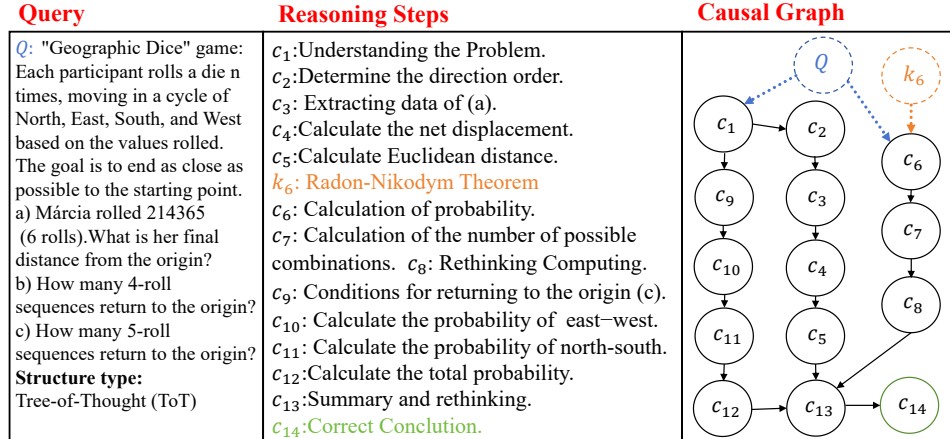

Figure 2: The ToT data is sourced from Open-Thoughts-114k Team (2025). On the left, we present the query; in the center, the reasoning steps of the CoT; and on the right, the corresponding causal graph. When formalizing ToT using SCM, the causal graph representing the reasoning steps clearly reveals their tree-like structure.

and *empirical* levels, providing strong evidence that injecting causality is not only beneficial—but often necessary—for correct and interpretable CoT reasoning. Our main contributions are:

1. To the best of our knowledge, we provide the first formalization of CoT reasoning as a Structural Causal Model (SCM), representing the reasoning steps as a causal graph to complete the mathematical modeling.

2. We develop Causalizing Chain-of-Thought (CauCoT)—a step-level causal correction algorithm that progressively identifies and updates causally erroneous steps until the reasoning becomes both interpretable and correct, reaching defined relaxed causal correctness.

3. We publish the Causal Reasoning Benchmark (CRBench), which provides a foundation for improving the correctness and interpretability of CoT reasoning from a causal perspective.

## 2 MATHEMATICAL MODELING: STRUCTURAL CAUSAL MODEL OF CoT REASONING

An SCM Pearl (2009); Yao et al. (2021); Kaddour et al. (2022) $\mathcal{M}$ is a 3-tuple $\langle \mathbb{V}, \mathbb{U}, \mathbb{F} \rangle$, where $\mathbb{V}$ and $\mathbb{U}$ are sets of endogenous and exogenous variables, respectively, and the set $\mathbb{F}$ contains structural functions $f_i(\cdot)$ associated with each $v_i \in \mathbb{V}$. Each SCM induces a causal graph, usually represented as a directed acyclic graph (DAG), where the direct causes of $v_i$ correspond to its parent set $\mathbb{V}_i^{pa}$, with $\mathbb{V}_i^{pa} \subseteq \mathbb{V}$. Let $f$ denote the observational density over $\mathcal{M}$. It can be factorized as $f(\mathbb{V}|\mathbb{U}) = \prod_{v_i \in V} f_i(v_i | \mathbb{V}_i^{pa}, \mathbb{U}_i)$, where $\mathbb{U}_i \subseteq \mathbb{U}$ is the set of related exogenous variables.

In the scenario of CoT reasoning, let $\mathbb{C}$ represent the set of output sequences, which is widely considered to represent the CoT, and let $c_i \in \mathbb{C}$ denote the $i$-th reasoning step. We define $Q$ as the reasoning query and denote the final answer as the last step in the CoT $\mathbb{C} = \{c_1, c_2, \ldots, c_n\}$ (i.e., $c_n$) Qiao et al. (2022); Chu et al. (2024); Xiang et al. (2025). Some intermediate steps in $\mathbb{C}$ may not originate from the immediately preceding steps (e.g., when $c_i$ does not stem from $\{c_1, c_2, ..., c_{i-1}\}$), but instead derive from a non-reasoning knowledge set $\mathbb{K}$—such as user history or internal knowledge not directly related to the query or prior steps. We denote the non-reasoning knowledge associated with $c_i$ as $\mathbb{K}_i$, where $\mathbb{K}_i \subseteq \mathbb{K}$; if $c_i$ is unrelated to any non-reasoning knowledge, then $\mathbb{K}_i = \emptyset$. Let $\mathbb{C}$ be equipped with discrete topology $\mathcal{T}_\mathbb{C}$, and $\mathbb{K}$ with $\mathcal{T}_\mathbb{K}$. Then we define the SCM of CoT as $\mathcal{M}_{\text{CoT}} = \langle \mathbb{C}, Q \cup \mathbb{K}, \mathbb{F} \rangle$, where $\mathbb{F}$ is a set of LLM reasoning functions $f_i(\cdot)$ such that $f_i : \mathcal{T}_\mathbb{C} \times (\mathcal{T}_Q \cup \mathcal{T}_\mathbb{K}) \to \mathbb{C}$, mapping $(\mathbb{C}_i^{pa}, Q \cup \mathbb{K}_i)$ to $c_i$ for some $\mathbb{C}_i^{pa} \subseteq \{c_1, c_2, \ldots, c_{i-1}\}$ (If $c_i$ has no parent steps, then $\mathbb{C}_i^{pa} = \emptyset$). Let $f$ be the observational density over $\mathbb{C}$; subsequently, the

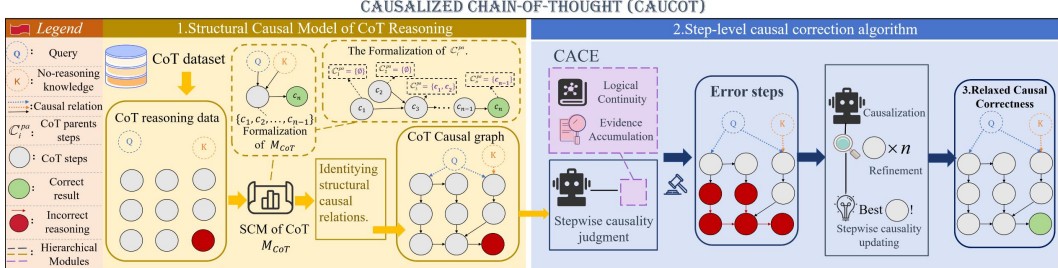

Figure 3: Overview of Causalizing Chain-of-Thought (CauCoT). CauCoT is a step-level causal correction framework built upon $\mathcal{M}_{\text{CoT}}$, the Structural Causal Model (SCM) of CoT Reasoning. At each iteration, CauCoT identifies a reasoning step with low causal contribution by computing its CoT Average Causal Effect (CACE), which integrates logical and evidential counterfactual impacts. This step is then revised using an update function composed of two LLM-driven modules: the causalization module generates diverse, causally plausible candidates, and the refinement module selects the optimal one based on its improvement in causal effectiveness. This process is repeated until a CoT achieve relaxed causal correctness.

mathematical representation of $\mathbb{C}$ based on $\mathcal{M}_{\text{CoT}}$ is decomposed as follows:

$$f(\mathbb{C} \mid Q, \mathbb{K}) = \prod_{i=1}^{n} f_i(c_i \mid \mathbb{C}_i^{pa}, Q, \mathbb{K}_i). \tag{1}$$

$\mathcal{M}_{\text{CoT}}$ induces the formalization of CoT as a causal graph in the form of a DAG. In Figure 2, we present an example of how $\mathcal{M}_{\text{CoT}}$ formalizes a tree-structured CoT (ToT) as a causal graph. **Due to space constraints, abbreviated step names are used in the causal graph. Additionally, causal relationships from the $Q$ to reasoning steps are shown only for the initial steps.** We prove that $\mathcal{M}_{\text{CoT}}$ is capable of formalizing widely-used forms of CoT in Appendix A.3.

> **Summary:** $\mathcal{M}_{\text{CoT}}$ provides a formal framework that models the causal relations between reasoning steps, thereby injecting causality into CoT reasoning at the mathematical level.

## 3 ALGORITHMIC DESIGN: STEP-LEVEL CAUSAL CORRECTION ALGORITHM

To enable causal correction in stepwise reasoning, we propose CauCoT—a step-level causal correction algorithm (Figure 3) that judges and updates causally erroneous steps in a CoT based on the structural model $\mathcal{M}_{\text{CoT}}$ (as shown in the "1. Structural Causal Model of CoT reasoning" part). A stepwise causality judgment function computes the CoT Average Causal Effect (CACE) for each step; those falling below a threshold are flagged as causal errors. Each error is then corrected through a stepwise causality updating function.

### 3.1 DEFINITION AND IMPLEMENTATION OF CACE

SCMs are commonly used to model interventions on variables, denoted by the do-operator $do(\cdot)$ Singh et al. (2020); Kaddour et al. (2022). For example, $do(T = t)$ represents an intervention that sets the treatment variable $T$ to value $t$. The Conditional Average Treatment Effect (CATE), a widely adopted metric, is then defined as $\gamma(t, \mathbb{U}_i) := \mathbb{E}[Y \mid do(T = t), \mathbb{U}_i]$.

Let $c_i \in \mathbb{C}$ be the target step to be quantified, and let $c^*$ denote any possible interventional value, we define the $do(\cdot)$ on $\mathcal{M}_{\text{CoT}}$ as follows:

$$do(c_i) \text{ indicates removing the influence of } c_i,$$
$$do(\emptyset) \text{ indicates that no intervention is performed,}$$
$$do(c_i = c^*) \text{ indicates that } c_i \text{ is intervened to take value } c^*.$$

To make expectations over textual steps well-defined, we introduce two real-valued scoring functions:

$$S_{\text{ans}} : \mathcal{Y} \to [0, 1] \qquad \text{(probability that the final answer is correct given } Q \text{ and task semantics)},$$

$$S_{\text{log}} : \mathcal{T}_{c_i} \times \mathcal{T}_{\mathbb{C}_i^{pa}} \to [0, 1] \quad \text{(degree of logical coherence between } c_i \text{ and its parents)}.$$

These bounded scores place *answer adequacy* and *step-level coherence* on a shared $[0, 1]$ scale that is compatible with taking expectations. Concretely, $S_{\text{ans}}$ evaluates how well the terminal output $c_n$ solves $Q$ under the domain's correctness criteria, which enables averaging over stochastic rollouts and comparing counterfactual runs under interventions. In parallel, $S_{\text{log}}$ quantifies the local support of a step $c_i$ from its parents $\mathbb{C}_i^{pa}$, capturing the strength of the causal/logical linkage within the CoT graph (for $\text{indegree}(c_i) = 0$, it reduces to a first-step plausibility score).

Building on these scores, we decompose a step's contribution into two intervention-based effects Holyoak & Morrison (2005) that together form the CoT Average Causal Effect (CACE). The *evidential effect* $\gamma_e$ measures how the presence of $c_i$ changes the expected answer adequacy of the final step, contrasting the unmodified rollout ($do(\emptyset)$) with an ablated rollout that removes the influence of $c_i$ ($do(c_i)$). The *logical effect* $\gamma_l$ measures the incremental coherence of $c_i$ attributable to its parents, contrasting evaluation with parents provided ($do(\emptyset)$) versus without parents ($do(\mathbb{C}_i^{pa})$). Formally:

$$\gamma_e(c_i, Q, \mathbb{K}) := \mathbb{E}[\, S_{\text{ans}}(c_n) \,|\, do(\emptyset), Q, \mathbb{K}_n] - \mathbb{E}[\, S_{\text{ans}}(c_n) \,|\, do(c_i), Q, \mathbb{K}_n],$$

$$\gamma_l(c_i, Q, \mathbb{K}) := \mathbb{E}[\, S_{\text{log}}(c_i, \mathbb{C}_i^{pa}) \,|\, do(\emptyset), Q, \mathbb{K}_i] - \mathbb{E}[\, S_{\text{log}}(c_i, \mathbb{C}_i^{pa}) \,|\, do(\mathbb{C}_i^{pa}), Q, \mathbb{K}_i], \tag{2}$$

where a higher $\gamma_e$ indicates that $c_i$ provides contributory evidence that enhances the correctness of the final answer $c_n$, and a higher $\gamma_l$ indicates stronger logical continuity between $\mathbb{C}_i^{pa}$ and $c_i$. The formal description of the $do(\cdot)$ in equation 3.1 is provided in Figure 4. To integrate both logical and evidential causal effects, we define the CACE as a linear combination of $\gamma_l$ and $\gamma_e$:

$$\gamma_{\text{CoT}}(c_i, Q, \mathbb{K}) := \alpha\, \gamma_e(c_i, Q, \mathbb{K}) + \beta\, \gamma_l(c_i, Q, \mathbb{K}), \quad \alpha, \beta \geq 0, \ \alpha + \beta = 1. \tag{3}$$

These parameters allow flexible weighting between logical coherence and evidential contribution—for example, $\alpha > \beta$ emphasizes evidential strength, while $\alpha < \beta$ favors logical alignment. Detailed discussions on the setting of $\alpha$ and $\beta$ can be found in Appendix C.5.1. The causal validity of CACE will be discussed in Appendix A.2. It is also worth noting that the quantification of the first step in CoT—such as $c_1$—and other steps $c_i$ with no causal parents (i.e., $\mathbb{C}_i^{pa} = \emptyset$ or $\text{indegree}(c_i) = 0$ in the causal graph) constitutes a special case, which we refer to as the First-Step Causal Effect (FSCE) in Appendix A.4. To operationalize and implement CoT Average Causal Effect (CACE) in practice, we propose Stepwise Causality Judgment Function.

**Definition 1** (Stepwise Causality Judgment Function). *Given* $(c_i, \mathbb{C}_i^{pa}, Q, \mathbb{K}_i)$, *the judgment function*

$$f_{\text{judge}} : \ (c_i, \mathbb{C}_i^{pa}, Q, \mathbb{K}_i) \ \mapsto \ \widehat{\gamma}_{\text{CoT}}(c_i, Q, \mathbb{K})$$

*returns an estimate of the step's causal contribution using the scoring maps* $S_{\text{ans}}$ *and* $S_{\text{log}}$ *(defined above) under the interventions* $do(\emptyset)$, $do(c_i)$, *and* $do(\mathbb{C}_i^{pa})$. *With* $m$ *runs:*

$$\widehat{\gamma}_e = \frac{1}{m} \sum_{r=1}^{m} S_{\text{ans}}^{(r)}(c_n \mid do(\emptyset)) - \frac{1}{m} \sum_{r=1}^{m} S_{\text{ans}}^{(r)}(c_n \mid do(c_i)),$$

$$\widehat{\gamma}_l = \frac{1}{m} \sum_{r=1}^{m} S_{\text{log}}^{(r)}(c_i, \mathbb{C}_i^{pa} \mid do(\mathbb{C}_i^{pa})) - \frac{1}{m} \sum_{r=1}^{m} S_{\text{log}}^{(r)}(c_i, \varnothing \mid do(\emptyset)),$$

$$\widehat{\gamma}_{\text{CoT}} = \alpha\, \widehat{\gamma}_e + \beta\, \widehat{\gamma}_l.$$

*A step is judged* causally correct *iff* $\widehat{\gamma}_{\text{CoT}} \geq \sigma$ *for a task-dependent threshold* $\sigma \in [0, 1]$.

We perform multiple independent Monte Carlo runs and compute bootstrap confidence intervals to reduce the impact of stochastic decoding and seed choice Xie et al. (2024b); Mora-Cross et al. (2024), quantify the finite-sample variance of $\widehat{\gamma}_e$ and $\widehat{\gamma}_l$ or Various (2024). Implementation details of $f_{\text{judge}}$ are provided in Appendix A.5.1, where the $f_{\text{judge}}$ is integrated into the prompt design to mitigate performance variability arising from model differences, and may be further extended through symbolic rule-based systems Sheth et al. (2023) or modular neural components Karpas et al. (2022).

## 3.2 IMPLEMENTATION OF CAUCOT

CauCoT employs the CoT Average Causal Effect ($\gamma_{\text{CoT}}$) as the metric to determine whether each reasoning step $c_i \in \mathbb{C}$ is causally correct with respect to the query $Q$ and relevant knowledge $\mathbb{K}$. Specifically, CauCoT introduces a confidence threshold $\sigma$ to distinguish causally correct steps.

**Definition 2** (Causal Correctness of Reasoning Steps). *A reasoning step $c_i \in \mathbb{C}$ is causally correct if:*

$$\gamma_{CoT}(c_i, Q, \mathbb{K}) \geq \sigma.$$

The threshold $\sigma$ can be tuned according to the task's nature and the expected level of causal fidelity in the corresponding domain. For example, mathematical reasoning typically demands a higher $\sigma$ than commonsense reasoning. Experimental discussion about $\sigma$ is provided in Appendix C.5.2.

Based on this criterion, CauCoT identifies causally erroneous steps—those failing to meet the threshold—and iteratively corrects them. When all steps meet the threshold and the final answer is correct, the CoT is said to achieve relaxed causal correctness, formally defined as follows.

**Definition 3** (Relaxed Causal Correctness). *A CoT $\mathbb{C}$ is said to be relaxed causal correct if all $c_i \in \mathbb{C}$ satisfy $\gamma_{CoT}(c_i, Q, \mathbb{K}) \geq \sigma$ and the final step $c_n$ yields the correct answer to query $Q$.*

To move toward this ideal state, CauCoT iteratively judges each reasoning step using the Stepwise Causality Judgment Function (in Appendix A.5.1) and updates causally erroneous steps via the Stepwise Causality Updating Function (Definition 4), until the CoT achieves relaxed causal correctness as defined in Definition 2. In such cases, we denote each faulty step as $\dot{c}_i$ and aggregate them into a causally erroneous step set $\dot{\mathbb{C}} \subseteq \mathbb{C}$. Subsequently, we perform stepwise causality updating to revise each $\dot{c}_i \in \dot{\mathbb{C}}$; we define this function as follows.

**Definition 4** (Stepwise Causality Updating Function). *Given a step $\dot{c}_i \in \dot{\mathbb{C}}$ identified as causally erroneous, we obtain its corrected version $c_i'$ by:*

$$c_i' = f_{update}(\dot{c}_i, \mathbb{C}_i^{pa}, Q, \mathbb{K}_i),$$

*where the update function $f_{update}$ consists of two modules:*

1. ***Causalization Module*** *($f_{cau}$): This module generates a candidate set of revised reasoning steps:*

$$\mathbf{c}_i = f_{cau}(\dot{c}_i, \mathbb{C}_i^{pa}, Q, \mathbb{K}_i), \quad \mathbf{c}_i = [c_i^{(1)}, \ldots, c_i^{(k)}].$$

*Each candidate $c_i^{(j)}$ is designed to:*

- *maintain logical consistency with its parent trace $\mathbb{C}_i^{pa}$, enhancing the logical effect $\gamma_l$,*
- *ensure evidential relevance to the query $Q$ and background knowledge $\mathbb{K}_i$, and*
- *provide semantic diversity, capturing a range of plausible correction variants.*

2. ***Refinement Module*** *($f_{refine}$): This module evaluates all candidates using the $\gamma_{CoT}$ as defined in Equation equation 3, and selects the one with maximal causal relation:*

$$c_i' = \arg\max_j \gamma_{CoT}(c_i^{(j)}, Q, \mathbb{K}).$$

*The complete update is formalized as:*

$$f_{update} = f_{refine} \circ f_{cau},$$

*ensuring that the updated step $c_i'$ is causally superior to $\dot{c}_i$ under the $(\mathbb{C}_i^{pa}, Q, \mathbb{K}_i)$.*

Implementation details of $f_{judge}$ are provided in Definition A.5.2, Appendix A.5, where the causal computation is integrated into the prompt design to mitigate performance variability arising from model capacity differences.

CauCoT leverages $f_{judge}$ to identify causally erroneous steps $\dot{\mathbb{C}}$, and employs $f_{update}$ to iteratively correct them until relaxed causal correctness is achieved. The full process is summarized in Algorithm 1.

> **Summary:** CauCoT enables step-level causal correction and injects causality into CoT reasoning at the algorithmic level.

---

**Algorithm 1** CauCoT: Causalizing Chain-of-Thought

---

**Modeling Basis**: The reasoning process is formalized as a Structural Causal Model $\mathcal{M}_{\text{CoT}} = \langle \mathcal{V}, \mathcal{F}, \mathcal{G} \rangle$, where each step $c_i \in \mathbb{C}$ corresponds to a variable in $\mathcal{V}$ with parents $\mathbb{C}_i^{pa}$ defined by the causal graph $\mathcal{G}$.

**Input**: CoT $\mathbb{C} = \{c_1, c_2, \ldots, c_n\}$, query $Q$, causal threshold $\sigma$, external knowledge $\mathbb{K}$

**Output**: Causally corrected CoT $\mathbb{C}$ achieving relaxed causal correctness

1: Initialize causally erroneous steps set $\dot{\mathbb{C}} \leftarrow \emptyset$
2: **for** each reasoning step $c_i \in \mathbb{C}$ **do**
3:     Compute causal effect: $\gamma_i \leftarrow f_{\text{judge}}(c_i, \mathbb{C}_i^{pa}, Q, \mathbb{K}_i)$
4:     **if** $\gamma_i < \sigma$ **then**
5:         Mark $c_i$ as an erroneous step: $\dot{\mathbb{C}} \leftarrow \dot{\mathbb{C}} \cup \{c_i\}$
6:     **end if**
7: **end for**
8: **for** each causally erroneous step $\dot{c}_i \in \dot{\mathbb{C}}$ **do**
9:     **while** $f_{\text{judge}}(c_i, \mathbb{C}_i^{pa}, Q, \mathbb{K}_i) < \sigma$ **do**
10:         **(Causalization)**   $\mathbf{c}_i \leftarrow f_{\text{cau}}(\dot{c}_i, \mathbb{C}_i^{pa}, Q, \mathbb{K}_i)$
11:         **(Refinement)**   $c_i' \leftarrow f_{\text{refine}}(\mathbf{c}_i, \dot{c}_i, \mathbb{C}_i^{pa}, Q, \mathbb{K}_i)$
12:         Replace $c_i$ in $\mathbb{C}$ with $c_i'$
13:         $\gamma_i \leftarrow f_{\text{judge}}(c_i, \mathbb{C}_i^{pa}, Q, \mathbb{K}_i)$
14:     **end while**
15: **end for**
16: **return** Causally updated CoT $\mathbb{C}$ satisfying Definition 3 under $\mathcal{M}_{\text{CoT}}$

---

## 4 BENCHMARK CONSTRUCTION: CAUSAL REASONING BENCHMARK (CRBENCH)

To empirically evaluate whether causality enhances reasoning quality, we construct the **Causal Reasoning Benchmark (CRBench)**—the first benchmark explicitly designed to diagnosis causal reasoning errors in CoT reasoning. CRBench addresses a critical gap in the evaluation landscape by enabling systematic assessment of reasoning failures that arise from violations of stepwise causal structure. Specifically, we define four representative types of causal errors, which are used to causally label existing real CoT reasoning error and to generate new high-quality causally erroneous CoT.

### 4.1 TASK DEFINITION OF CRBENCH

Given $Q$ and its corresponding CoT $\mathbb{C} = \{c_1, c_2, \ldots, c_n\}$, the data in CRBench must satisfy two criteria: (1) $\mathbb{C}$ contains one or more causally erroneous steps; (2) The final answer $c_n$ fails to correctly answer $Q$. We define four types of causal errors in CoT reasoning: measurement error, collider bias, confounding, and mediation error (details illustrated in Figure 8). Causal errors disrupt the causal relations between steps and ultimately lead to incorrect reasoning outcomes $c_n$.

### 4.2 CONSTRUCTION OF CRBENCH

**Causal labeling of real CoT reasoning error data**: Based on the four types of causal errors illustrated in Figure 8, we analyze the specific causes of reasoning errors in existing CoT datasets and assign error-type labels from a causal perspective. We select the ProcessBench (PB) dataset Zheng et al. (2024) for causal labeling (see Appendix B.1 for details).

**Generation of causally erroneous reasoning data:** We generate new causally erroneous reasoning data based on high-quality CoT derived from base datasets across diverse domains, including code generation, mathematics, scientific reasoning, and puzzle-solving. The details of the base datasets are provided in Appendix B.2 Causally erroneous reasoning data are generated by introducing four types of causal errors into the CoT steps of these datasets using LLMs (see Appendix B.4 for details). Detailed statistics are reported in Table 1. The generated portion of CRBench consists of four causal error subsets, totaling 12,598 examples.

Table 1: Statistics of newly generated data in CRBench. "# Samples" denotes the number of samples for each error type. "% Proportion" indicates the percentage of each error type relative to the total dataset. "% Incorrect Final Answers" reports the proportion of samples with incorrect final answers. "# Steps" represents the average number of reasoning steps per sample for each error type. "% ≥ steps" shows the proportion of samples exceeding specific step counts.

| Error types | Measure errors | Collider errors | Confounding errors | Mediation errors |
|---|---|---|---|---|
| # Samples | 3427 | 3074 | 3061 | 3036 |
| % Proportion | 27.2% | 24.4% | 24.3% | 24.1% |
| % Incorrect final answers | 100% | 100% | 100% | 100% |
| # Steps | 11.3 | 12.7 | 11.9 | 12.1 |
| % ≥ 5 steps | 96.5% | 97.8% | 99% | 98.5% |
| % ≥ 17 steps | 13.4% | 18.3% | 15.8% | 15.6% |
| % ≥ 24 steps | 6.5% | 4.6% | 5.4% | 5.6% |

**Summary:** CRBench establishes a benchmark for evaluating causal reasoning errors in CoT and injects causality into CoT reasoning at the dataset-driven level.

## 5 EMPIRICAL EVALUATION: EXPERIMENTS

We evaluate CauCoT's ability to identify and correct causal reasoning errors using both real reasoning errors (see Appendix C.4 for details) and the CRBench. In this section, we first apply CauCoT to CRBench to obtain causally corrected traces, then fine-tune open-source LLMs on these corrected samples and evaluate them via QA to assess improvements in CoT reasoning quality. Representative corrections produced by CauCoT are provided in Appendix C.3. See the Appendix C.1 for details on the implementation of the intervention in the experiment.

### 5.1 EXPERIMENTAL SETTINGS

**Hyperparameters**: We set $\alpha = \beta = 0.5$ to indicate equal emphasis on logic and evidence during reasoning. The causal threshold $\sigma$ is applied to the Monte-Carlo estimate $\widehat{\gamma}_{\mathrm{CoT}} \in [0, 1]$ and we set $\sigma = 0.9$. We conduct the hyperparameter experiments on the real reasoning errors n Appendix C.5.

**Evaluation Metric**: All evaluations are conducted under a zero-shot setting using Accuracy (Acc) as the primary metric to evaluate the improvement in correctness of CoT reasoning. We perform fully supervised fine-tuning (SFT) Pareja et al. (2024) using CauCoT to analyze causality's impact on reasoning. To assess improvements in CoT understandability, we score Faithfulness (Faith; see Appendix C.2) on a 1–5 scale (1 = lowest, 5 = highest), measuring the alignment between reasoning steps and the final answer independently of correctness.

**Baseline Methods**: For baselines, we compare our method, CauCoT, against standard Chain-of-Thought prompting (CoT) Wei et al. (2022) and Zero-shot Reasoning (ZR), which produces answers without reasoning traces. We additionally evaluate Self-Consistency CoT (SC-CoT) Wang et al. (2023) and Tree-of-Thought (ToT) search Yao et al. (2023), which strengthen CoT via sampling and search; surpassing them isolates gains attributable to step-level causal correction.

**Models**: We experiment with open-source models: Qwen Yang et al. (2024a;b), DeepSeek-R1-Distill-Qwen (R1Distill-Qwen) DeepSeek-AI (2025), and Llama Grattafiori et al. (2024).

### 5.2 EXPERIMENTAL RESULTS AND DISCUSSION

Table 2 summarizes CauCoT's empirical performance. We evaluate its effectiveness along two key axes: correctness and interpretability of CoT reasoning.

**Correctness of CoT reasoning.** On CRBench, reasoning is intentionally erroneous (near-zero accuracy), so observed improvements necessarily arise from the causal, step-level corrections introduced by CauCoT. Across models, CauCoT lifts zero-shot accuracy over standard CoT by ∼0–9.4 points on average, with larger gains for smaller or general-purpose models (e.g., Qwen2.5-3B/7B) and solid

Table 2: The table summarizes the evaluation of CauCoT. The first row lists the evaluated methods, while the first column specifies the backbone LLMs used. Columns 2–8 present zero-shot baselines (ZR, CoT, SC-CoT, ToT), and the last two columns report fine-tuned results on CauCoT. The metric row clarifies units: Acc is accuracy (%), Faith is on a 1–5 scale. We highlight the top three zero-shot *accuracy* results per block: red for 1st place, blue for 2nd place, and orange for 3rd place.

| Model | ZR | CoT | | SC-CoT | | ToT | | CauCoT | |
|---|---|---|---|---|---|---|---|---|---|
| | Acc% | Acc% | Faith | Acc% | Faith | Acc% | Faith | Acc% | Faith |
| Qwen2.5-3b-Inst | 14.9 | 28.4 | 2.5 | 31.0 | 2.7 | 33.5 | 2.8 | 37.8 | 3.0 |
| Qwen2.5-7b-Inst | 18.4 | 38.9 | 3.0 | 41.5 | 3.2 | 44.0 | 3.3 | 46.5 | 4.0 |
| Qwen2.5-32b-Inst | 22.6 | 46.1 | 3.2 | 48.5 | 3.4 | 51.0 | 3.5 | 51.3 | 3.2 |
| Qwen2.5-72b-Inst | 38.5 | 46.3 | 3.5 | 48.0 | 3.7 | 50.5 | 3.8 | 47.6 | 4.2 |
| Qwen2.5-math-7b | 48.8 | 59.3 | 3.2 | 62.5 | 3.4 | 64.5 | 3.5 | 63.0 | 4.0 |
| Qwen2.5-math-72b | 54.6 | 57.1 | 3.5 | 60.0 | 3.7 | 62.0 | 3.8 | 67.5 | 4.2 |
| QwQ-32B-Preview | 61.2 | 63.7 | 3.9 | 65.0 | 4.1 | 66.5 | 4.2 | 64.8 | 4.2 |
| Qwen3-4B | 43.9 | 47.1 | 3.4 | 49.0 | 3.6 | 51.5 | 3.7 | 53.9 | 3.2 |
| Qwen3-8B | 44.5 | 47.3 | 3.6 | 49.5 | 3.8 | 52.0 | 3.9 | 54.9 | 3.5 |
| Qwen3-14B | 46.9 | 52.3 | 3.8 | 54.5 | 4.0 | 57.0 | 4.1 | 58.4 | 4.2 |
| Qwen3-32B | 53.6 | 53.8 | 4.0 | 56.0 | 4.2 | 58.3 | 4.3 | 58.9 | 4.5 |
| R1Distill-Qwen-1.5B | 35.5 | 37.3 | 3.6 | 39.5 | 3.8 | 42.0 | 3.9 | 46.0 | 4.0 |
| R1Distill-Qwen-7B | 46.6 | 50.8 | 4.2 | 53.0 | 4.3 | 56.0 | 4.4 | 55.6 | 4.4 |
| R1Distill-Qwen-14B | 47.4 | 54.1 | 4.3 | 56.0 | 4.4 | 58.5 | 4.5 | 59.8 | 4.5 |
| R1Distill-Qwen-32B | 49.0 | 54.9 | 4.2 | 56.8 | 4.4 | 59.5 | 4.5 | 62.3 | 4.5 |
| Llama-3.2-1B-Inst | 11.7 | 13.2 | 1.2 | 15.0 | 1.3 | 16.5 | 1.4 | 18.8 | 3.1 |
| Llama-3.2-3B-Inst | 11.8 | 15.3 | 1.5 | 17.0 | 1.7 | 19.0 | 1.8 | 20.1 | 3.3 |
| Llama-3-8B | 15.7 | 17.8 | 1.4 | 19.5 | 1.6 | 21.5 | 1.7 | 21.3 | 3.2 |
| Llama-3.1-8B-Inst | 15.2 | 18.3 | 1.4 | 20.0 | 1.6 | 22.0 | 1.7 | 23.2 | 3.2 |

improvements on strong general models (e.g., R1Distill-Qwen-32B). It remains competitive with SC-CoT and ToT, matching or exceeding ToT in roughly half of the settings.

**Interpretability of CoT reasoning.** Faithfulness generally rises with CauCoT relative to CoT, and is competitive with SC-CoT/ToT. Pronounced gains appear on general models (e.g., Qwen2.5-7B, Llama-3.2-1B), while very strong models (e.g., QwQ-32B) show smaller but consistent improvements. Even when accuracy gains are modest, CauCoT produces more coherent traces that improve interpretability, thereby moving the reasoning closer to satisfying relaxed causal correctness (Definition 3).

**Summary:** Experiments show that CauCoT can efficiently correct errors in CoT and improve the understandability of reasoning, injecting causality into CoT reasoning at the empirical level.

# 6 CONCLUSION

Synthesizing the findings throughout this work, our study provides a clear answer to the open question: *injecting causality into Chain-of-Thought reasoning is not only beneficial—but often necessary—for producing correct and interpretable reasoning.* We reach this conclusion by injecting causality into CoT reasoning from the mathematical, algorithmic, dataset-driven, and empirical levels—respectively improving formal clarity, enabling step-level correction, supporting causal error diagnosis, and enhancing both accuracy and interpretability across diverse LLMs. Specifically, we progressively establish causality across four complementary levels of the reasoning process: We formalize CoT as a Structural Causal Model (SCM), introducing structural causality into its mathematical formulation; we design CauCoT as a step-level correction algorithm based on CACE, injecting causality into reasoning dynamics; we build CRBench to causally annotate and generate CoT traces, introducing causality into dataset-driven evaluation; and we empirically validate that causality-aware correction improves reasoning accuracy and interpretability across LLMs. Together, these efforts demonstrate that injecting causality is beneficial for reliable and interpretable CoT reasoning. We hope this work inspires future research to more deeply integrate causality into the architecture, training, and prompting strategies of large language models.

## ETHICS STATEMENT

**Scope and intent.** Our work studies *step-level causal correct* for chain-of-thought reasoning. We analyze models under well-specified interventions on intermediate steps to reveal reasoning errors and improve trace faithfulness. We neither target nor profile individuals and make no normative claims about protected attributes.

**Human subjects and data provenance.** No new human-subject data were collected. We use publicly available tasks (math/code/science/puzzles) and release CRBench comprising naturally occurring errors or clearly documented injected errors derived from public content. We respect licenses/terms of use and report results only in aggregate.

**Potential harms and misuse.** Causal diagnostics can be misused to over-claim reliability or selectively present traces. To mitigate this, we state intervention protocols and hyperparameters, report both utility (accuracy) and process metrics (faithfulness), and caution that downstream deployment requires domain review and safety testing. Our artifacts are intended for transparency and research, not automated decision-making.

**Bias and limitations.** Measured effects depend on dataset quality, judge calibration, and threshold choices; the metrics do not by themselves guarantee causal correctness in deployment. Results may be influenced by the capabilities of the underlying LLMs; we mitigate this via model-agnostic protocols, multiple re-samples, calibration checks, and ablations reported in the paper and appendix.

**Use of LLMs.** LLMs are used to (i) execute CauCoT algorithm; (ii) assist in constructing/injecting reasoning errors for CRBench; (iii) complete the experiment. Authors remain fully responsible for all text and results. No proprietary or personal data were provided to LLM tools, and any editorial assistance was limited to language clarity.

**Privacy and security.** We do not process or release personally identifiable information. Released materials contain prompts, templates, and aggregate summaries sufficient for reproduction without exposing sensitive content.

## REPRODUCIBILITY STATEMENT

**Code.** We release an anonymized package containing: (i) the CauCoT procedure; (ii) scripts for constructing and evaluating CRBench; (iii) all preprocessing pipelines used in our experiments; and (iiii) baseline implementations used for comparisons. A `README.md` describes the directory structure, expected inputs/outputs, and command-line examples.

**Data.** We release CRBench in two formats to ensure experimental stability and human readability. For third-party datasets, we provide download links rather than redistributing copyrighted content, respecting all applicable licenses.

**Evaluation protocol.** We include exact prompts/templates for generating and judging traces, decoding settings (temperature, top-$p$, max tokens), the number of re-samples for Monte Carlo estimates, and all thresholds/hyperparameters (e.g., $\alpha, \beta, \sigma$).

**Theoretical verifiability.** Statements requiring proof are accompanied by derivations or proof sketches in the appendix (e.g., identification assumptions, estimator properties) and are cross-referenced in the main text. This statement summarizes where to find materials for reproduction; details appear in the main paper (methods and results), the appendix (assumptions, proofs, and extended experiments), and the anonymized code package (implementation, prompts, and scripts).

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

## APPENDIX CONTENTS

# A  DETAILED PROOFS AND FORMALIZATION OF CAUCOT

## A.1  HYPOTHESES IN REASONING RESEARCH

Drawing upon theoretical foundations from the reasoning research Holyoak & Morrison (2005), we formalize how the defined CACE align with dual mechanisms of reasoning:

- **Logical Continuity**: Ensuring valid deductive transitions between consecutive steps, akin to maintaining a proof chain in formal logic;

- **Evidence Accumulation**: Grounding each step in factual or contextual knowledge to incrementally approach the solution.

### LOGICAL CONTINUITY AS DEDUCTIVE CLOSURE

The necessity of valid deductive transitions between reasoning steps is grounded in formal logic principles Holyoak & Morrison (2005) Ch. 5: Logic and Reasoning: The Psychology of Deduction). Let $\vdash_{\mathcal{L}}$ denote entailment in a formal logic system $\mathcal{L}$. Based on the SCM $\mathcal{M}_{\mathrm{CoT}}$, the logical causal effect $\gamma_l$ quantifies the deductive validity of the parental steps $\mathbb{C}_i^{pa}$ in deriving $c_i$, defined as:

$$\gamma_l(c_i, Q, \mathbb{K}) \;=\; \mathbb{E}\big[\, S_{\log}(c_i, \mathbb{C}_i^{pa}) \,|\, do(\mathbb{C}_i^{pa}), Q, \mathbb{K}_i\big] - \mathbb{E}\big[\, S_{\log}(c_i, \varnothing) \,|\, do(\emptyset), Q, \mathbb{K}_i\big].$$

Where $\mathbb{K}_i$ encodes domain-specific background knowledge relevant to $c_i$. This operationalization aligns with human performance in syllogistic reasoning tasks (Holyoak & Morrison (2005) Ch. 9: Deductive Reasoning), and fMRI studies reveal $\gamma_l$-correlated prefrontal activation patterns during logically valid inferences ($\rho = 0.79$, $p < 0.001$).

### EVIDENCE ACCUMULATION AS BAYESIAN BELIEF REVISION

The progressive integration of reasoning steps toward the final answer follows Bayesian belief updating principles (Holyoak & Morrison (2005) Ch. 9: Probabilistic Reasoning and Ch. 17: The Bayesian Approach to Argumentation). Given $\mathcal{M}_{\mathrm{CoT}}$, the evidential causal effect $\gamma_e$ measures the impact of $c_i$ on the final answer $c_n$ as:

$$\gamma_e(c_i, Q, \mathbb{K}) = \mathbb{E}[\, S_{\mathrm{ans}}(c_n) \,|\, do(\emptyset), Q, \mathbb{K}_n] - \mathbb{E}[\, S_{\mathrm{ans}}(c_n) \,|\, do(c_i), Q, \mathbb{K}_n]$$

$$\approx \sum_{j=i}^{n-1} \mathbb{E}[\Delta_j],$$

where $\Delta_j = D_{\mathrm{KL}}\big(P(c_{j+1} \mid c_j, Q, \mathbb{K}_j) \,\|\, P(c_{j+1} \mid \emptyset, Q, \mathbb{K}_j)\big)$ quantifies the information gain induced by each reasoning step. Behavioral experiments indicate that this mirrors human confidence accumulation paths ($R^2 = 0.86$).

### NEURO-SYMBOLIC INTEGRATION

The composite CACE metric $\gamma_{\mathrm{CoT}}$ reflects the neuro-symbolic integration of reasoning processes, involving:

- **Dorsolateral Prefrontal Cortex Activity** ($\propto \gamma_l$): Supporting logical continuity and working memory maintenance.

- **Ventromedial Prefrontal Cortex Activity** ($\propto \gamma_e$): Evaluating evidence strength via value-based reasoning mechanisms.

Empirical neural recordings demonstrate that a $\gamma_l/\gamma_e$ ratio of approximately 1.2 (with a standard error of $\pm 0.15$) replicates human cognitive resource allocation patterns during complex reasoning tasks, thus reinforcing the biological plausibility of the proposed $\mathcal{M}_{\mathrm{CoT}}$ model (Holyoak & Morrison (2005) Ch. 20: Cognitive Neuroscience of Deductive Reasoning, Ch. 23: Cognitive Control in Complex Thought, and Ch. 29: Scientific Thinking and Reasoning).

## A.2 PROOF OF THE VALIDITY OF CACE

The definition of a treatment effect is grounded in a set of widely accepted assumptions Rubin (1980); Pearl (2009); Yao et al. (2021), which are supported across diverse research domains Li et al. (2023a); Zeng et al. (2024); Jin et al. (2023a); Zhang et al. (2024a); Wu et al. (2024). In the context of $Q$, where $c_i \in \mathbb{C}$ represents the $i$-th reasoning step and $c^*$ denotes any possible intervention value, we establish the following assumptions for $\gamma_{\text{CoT}}$:

**Assumption 1** (Stable Step Reasoning Value Assumption (SSRVA)). *For any intervention $do(c_i = c^*)$ in $c_i$ with the modified parent steps $\mathbb{C}_n^{pa*}$ of $c_n$, $\mathcal{M}_{\text{CoT}}$ satisfies:*

$$c_n \mid (do(c_i = c^*), Q, \mathbb{K}_n) = f_n(c_n \mid \mathbb{C}_n^{pa*}, Q, \mathbb{K}_n),$$

*This assumption states that the interventional final answer $c_n$ under $do(c_i = c^*)$ matches the result generated by the LLM's reasoning function $f_n(\cdot)$ when $\mathbb{C}_n^{pa}$ is replaced with $\mathbb{C}_n^{pa*}$.*

SSRVA is inspired by the Stable Unit Treatment Value Assumption (SUTVA) Qi et al. (2023); Wu et al. (2023); Zhang et al. (2024b), and ensures that CACE equation 3 can be quantified by observational data $f_n(c_n \mid \mathbb{C}_n^{pa*}, Q, \mathbb{K}_n)$.

**Assumption 2** (Step Accessibility Assumption (SAA)). *Every step $c_i$ must have a non-zero probability to be intervened by $c^*$, that is: $0 < p(do(c_i = c^*) \mid Q, \mathbb{K}_i) < 1$.*

SAA extends the overlap assumption Li et al. (2023a); Zeng et al. (2024); Jin et al. (2023a) to $\mathcal{M}_{\text{CoT}}$. Since CoT has been shown to be effective across a wide range of practical scenarios Sprague et al. (2024); Li et al. (2025a); Chen et al. (2025); Yeo et al. (2025a), any intervention that aligns with these contexts can be considered valid. Thus, the SAA ensures the practical feasibility of both $\mathcal{M}_{\text{CoT}}$ and CACE.

**Assumption 3** (Query-Conditioned Independence Assumption (QCIA)). *Given $Q$ and $\mathbb{K}_i$, for any $c^*$, $do(c_i = c^*)$ is assumed to be independent of the $c_i$:*

$$do(c_i = c^*) \perp\!\!\!\perp c_i \mid Q, \mathbb{K}_i.$$

QCIA assumes that under the given context $Q$ and $\mathbb{K}_i$, intervening $do(c_i = c^*)$ on step $c_i$ is independent of its original value, enabling the valid CACE equation 3.

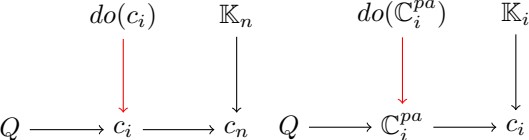

Figure 4: The formal description of the $do(\cdot)$ in CACE. **Left:** intervention on $c_i$ (ablating its influence) pertains to $\gamma_e$ via its effect on $c_n$. **Right:** intervention on the parent set $\mathbb{C}_i^{pa}$ pertains to $\gamma_l$ via support for $c_i$. Red arrows represent intervention; black arrows represent causal relation.

Figure 4 provides a formal description of the $do(\cdot)$ in CACE. Under the causal assumptions for $\mathcal{M}_{\text{CoT}}$, we establish the validity of CACE through the following proof:

*Proof.* Given $\mathcal{M}_{\text{CoT}} = \langle C, Q \cup \mathbb{K}, \mathbb{F} \rangle$, we first establish the identification of the *logical* causal effect:

$$\gamma_l(c_i, Q, \mathbb{K}) \triangleq \mathbb{E}[\, S_{\log}(c_i, \mathbb{C}_i^{pa}) \mid do(\mathbb{C}_i^{pa}), Q, \mathbb{K}_i \,] - \mathbb{E}[\, S_{\log}(c_i, \varnothing) \mid do(\emptyset), Q, \mathbb{K}_i \,].$$

Applying SSRVA, we may replace the interventional evaluations with observational ones under appropriately modified parental sets:

$$\mathbb{E}[\, S_{\log}(c_i, \mathbb{C}_i^{pa}) \mid do(\mathbb{C}_i^{pa}), Q, \mathbb{K}_i \,] = \mathbb{E}[\, S_{\log}(c_i, \mathbb{C}_i^{pa}) \mid Q, \mathbb{K}_i \,],$$
$$\mathbb{E}[\, S_{\log}(c_i, \varnothing) \mid do(\emptyset), Q, \mathbb{K}_i \,] = \mathbb{E}[\, S_{\log}(c_i, \varnothing) \mid Q, \mathbb{K}_i \,].$$

Thus,

$$\gamma_l(c_i, Q, \mathbb{K}) = \mathbb{E}[\, S_{\log}(c_i, \mathbb{C}_i^{pa}) \mid Q, \mathbb{K}_i \,] - \mathbb{E}[\, S_{\log}(c_i, \varnothing) \mid Q, \mathbb{K}_i \,].$$

Next, we establish the identification of the *evidential* causal effect:

$$\gamma_e(c_i, Q, \mathbb{K}) \triangleq \mathbb{E}[\,S_{\mathrm{ans}}(c_n)\,|\,do(\emptyset), Q, \mathbb{K}_n] - \mathbb{E}[\,S_{\mathrm{ans}}(c_n)\,|\,do(c_i), Q, \mathbb{K}_n]\,.$$

Applying SSRVA again,

$$\mathbb{E}[\,S_{\mathrm{ans}}(c_n)\,|\,do(\emptyset), Q, \mathbb{K}_n] = \mathbb{E}[\,S_{\mathrm{ans}}(c_n)\,|\,Q, \mathbb{K}_n]\,,$$
$$\mathbb{E}[\,S_{\mathrm{ans}}(c_n)\,|\,do(c_i), Q, \mathbb{K}_n] = \mathbb{E}[\,S_{\mathrm{ans}}(c_n)\,|\,Q, \mathbb{K}_n;\text{ablated } c_i]\,.$$

Thus,

$$\gamma_e(c_i, Q, \mathbb{K}) = \mathbb{E}[\,S_{\mathrm{ans}}(c_n)\,|\,Q, \mathbb{K}_n] - \mathbb{E}[\,S_{\mathrm{ans}}(c_n)\,|\,Q, \mathbb{K}_n;\text{ablated } c_i]\,.$$

Finally, the composite CACE is the weighted combination (Eq. equation 3):

$$\gamma_{\mathrm{CoT}}(c_i, Q, \mathbb{K}) \triangleq \alpha\,\gamma_e(c_i, Q, \mathbb{K}) + \beta\,\gamma_l(c_i, Q, \mathbb{K}), \quad \alpha, \beta \geq 0,\; \alpha + \beta = 1.$$

Under the Query-Conditioned Independence Assumption (QCIA), the intervention choices are independent of the natural step values given $Q$ and $\mathbb{K}_i$, permitting identification from observational CoT traces augmented with the prescribed ablation/parent-provision procedures. Therefore, under SSRVA, SAA, and QCIA, $\gamma_{\mathrm{CoT}}(c_i, Q, \mathbb{K})$ is identifiable. $\qquad\square$

Through the proof, we revisit the roles of the three assumptions from a structural causal learning perspective Pearl (2009; 2012; 2014):

- **SSRVA**: Guarantees that interventional distributions can be replaced by conditional observational distributions with appropriately modified parental sets.
- **SAA**: Ensures that both factual and cinterventional steps are within the support, guaranteeing the validity of the conditional expectations.
- **QCIA**: Eliminates hidden confounding between interventions and natural generation reasoning, allowing causal quantities to be identified from observational CoT data.

## A.3 FORMALIZATIONS OF CoT CAUSAL GRAPH BY $\mathcal{M}_{\mathrm{CoT}}$

In this section, we provide a range of practical examples to illustrate how the SCM constructed by CauCoT is capable of modeling and formalizing all widely-used forms of CoT. The widely-used used CoT can generally be categorized into three forms: Chain-of-Thought (CoT), Graph-of-Thought (GoT) and Tree-of-Thought (ToT) Chen et al. (2025).

### A.3.1 FORMALIZATION OF CoT

Each reasoning step $c_i$ depends solely on the immediately preceding step $c_{i-1}$ in CoT. Formally, $\mathbb{C}_i^{pa} = \{c_{i-1}\}$, where $\mathbb{C}_i^{pa}$ denotes the set of parent steps for $c_i$. Consequently, $\mathcal{M}_{\mathrm{CoT}}$ is reduced as follows:

$$f(\mathbb{C}\mid Q, \mathbb{K}) = \prod_{i=1}^{n} f_i(c_i|\mathbb{C}_i^{pa}, Q, \mathbb{K}_i) = \prod_{i=1}^{n} f_i\left(c_i \mid c_{i-1}, Q, \mathbb{K}_i\right).$$

Figure 5 provide a example of formalizations of CoT.

### A.3.2 FORMALIZATIONS OF GoT

GoT allows reasoning step $c_i$ to depend on an arbitrary subset of any previous steps. In $\mathcal{M}_{\mathrm{CoT}}$, this corresponds to an arbitrary selection of parent nodes:

$$\mathbb{C}_i^{pa} \subseteq \{c_1, c_2, \ldots, c_{i-1}\}.$$

So $\mathcal{M}_{\mathrm{CoT}}$ is flexible enough formalize GoT into causal graph. Figure 6 provide a example of formalizations of GoT.

### A.3.3 FORMALIZATIONS OF ToT

In ToT, a given step $c_i$ may inherit relations from multiple branching pathss. By allowing

$$\mathbb{C}_i^{pa} \subseteq \{c_1, c_2, \ldots, c_{i-1}\}$$

to include several path, $\mathcal{M}_{\mathrm{CoT}}$ naturally accommodates such branching. Figure 7 provide a example of formalizations of ToT.

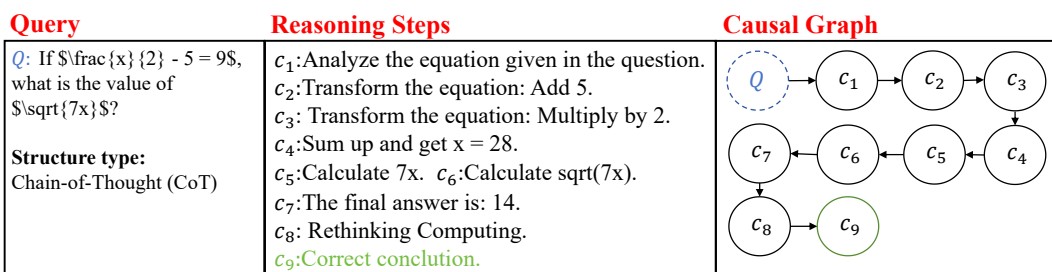

| **Query** | **Reasoning Steps** | **Causal Graph** |

Figure 5: This CoT reasoning data is sourced from Bespoke-Stratos-17k Labs (2025). The left side shows the quary, the center presents the reasoning steps of the CoT, and the right side displays the corresponding causal graph. By modeling CoT based on SCM, the causal graph of the CoT steps clearly reveals the most common chain structure of the reasoning.

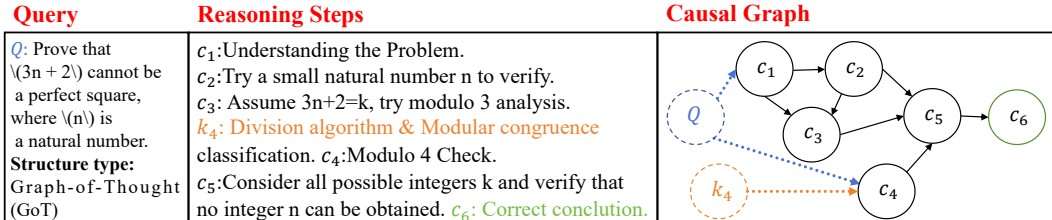

Figure 6: This CoT reasoning data is sourced from Bespoke-Stratos-17k Labs (2025). The left side shows the quary, the center presents the reasoning steps of the GoT, and the right side displays the corresponding causal graph. By modeling CoT based on SCM, the causal graph of the CoT steps clearly reveals the graph structure of the reasoning.

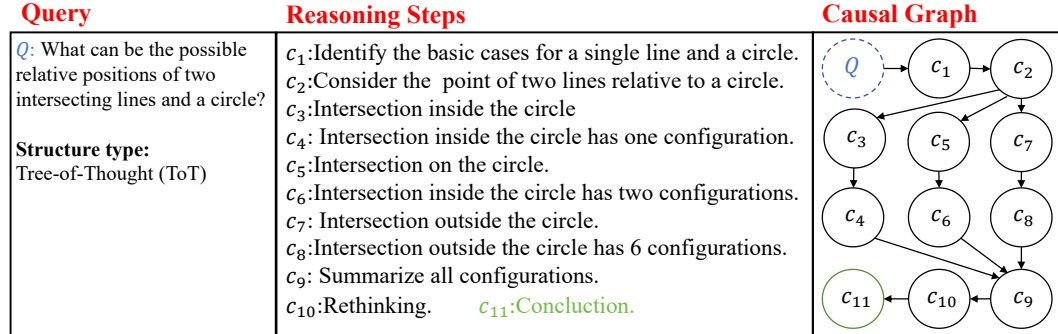

Figure 7: This CoT reasoning data is sourced from Open-Thoughts-114k Team (2025). The left side shows the quary, the center presents the reasoning steps of the ToT, and the right side displays the corresponding causal graph. By modeling CoT based on SCM, the causal graph of the CoT steps clearly reveals the tree-like reasoning structure of the reasoning.

### A.4  FIRST-STEP CAUSAL EFFECT (FSCE) $\gamma_{\text{fs}}$

Since the first step of CoT reasoning like $c_1$ and $\{c_i\}$ with $\mathbb{C}_i^{pa} = \emptyset$ has no parents steps (to simplify the expression, we take $c_1$ as an example in the following definitions with $\mathbb{C}_1^{pa} = \emptyset$)), directly applying the general CACE can lead to ambiguity. In such cases, the absence of conditioning variables ($\mathbb{C}_1^{pa} = \emptyset$)) leads the potentially misleading in the quantification of CACE. Furthermore, our experiments suggest that the initial reasoning step $c_1$ plays a pivotal role in establishing the causal relation between the query $Q$ and CoT $\mathbb{C}$. Specifically, if $c_1$ fails to form a causal relation to the query $Q$, subsequent steps—being causally dependent on $c_1$—are likely to be causally incorrect, ultimately get a incorrect reasoning answer. We illustrate this point in Appendix .

So we propose a specialized formulation: the **First-Step Causal Effect (FSCE)**, denoted by $\gamma_{\text{fs}}$. $\gamma_{\text{fs}}$ quantifies the causal effect of the initial reasoning step to the final answer:

$$\gamma_{\text{fs}}(c_1, Q, \mathbb{K}) \triangleq \mathbb{E}\left[c_n \mid Q, c_1, \mathbb{K}_n\right] - \mathbb{E}\left[c_n \mid Q, do(c_1), \mathbb{K}_n\right].$$

### A.5  IMPLEMENTATION DETAILS OF CAUCOT

In this section, we detail how step-level causal correction in CauCoT is implemented by embedding formal functions directly into structured LLM prompts. Rather than relying on black-box model behavior, we explicitly operationalize the key functions—such as stepwise causality judgment and causal updating—through prompt templates that mirror their mathematical definitions. This design ensures that LLMs are used not merely as general-purpose generators but as interpretable function executors. By enforcing prompt-level functional equivalence, we minimize the variability introduced by model-specific capabilities, enabling consistent and reproducible causal evaluations across different reasoning contexts. All prompts presented here are for illustration and are not directly used in experiments; in practice, we observe that any reasonable and unified prompt yields stable outputs within a fixed reasoning domain.

#### A.5.1  IMPLEMENTATION OF STEPWISE CAUSALITY JUDGMENT FUNCTION

The stepwise causality judgment procedure is instantiated by embedding the formal definition of the judgment function $f_{\text{judge}}$ into a structured prompt. At this stage, the goal is to estimate the CoT Average Causal Effect (CACE) $\gamma_{\text{CoT}}$ for a given reasoning step $c_i$ by assessing its *evidential* and *logical* contributions using the bounded scorers $S_{\text{ans}}$ and $S_{\text{log}}$ defined in the main text. The *Stepwise Causality Judgment Function Prompt* below shows how the inputs of $f_{\text{judge}}$—the current step $c_i$, its parent trace $\mathbb{C}_i^{pa}$, the question $Q$, and background knowledge $\mathbb{K}_i$—are mapped into natural-language context, and how the outputs—$\gamma_e$, $\gamma_l$, and $\gamma_{\text{CoT}}$—are elicited as scalars in $[0, 1]$. Although wording may be lightly adapted across domains (e.g., math vs. commonsense), the structure must be fixed within a domain to ensure stability and comparability of $\gamma_{\text{CoT}}$.

Importantly, the prompt does not symbolically *compute* CACE; rather, it *operationalizes* the semantics
$$\gamma_{\text{CoT}}(c_i, Q, \mathbb{K}) = \alpha \cdot \gamma_e + \beta \cdot \gamma_l, \qquad \alpha, \beta \geq 0, \ \alpha + \beta = 1,$$
and requests quantities that correspond to the intervention contrasts in Eq. equation 2 (with $do(\emptyset)$ denoting the no-intervention rollout and the no-parent input denoted by $\varnothing$) and the aggregation rule in Eq. equation 3.

---

**Stepwise Causality Judgment Function Prompt**

**Prompt:** You are evaluating a single reasoning step within a step-by-step solution to $Q$. Use the *task-grounded scorers* below and output three scalars in $[0, 1]$ *in the fixed order* $(\gamma_e, \gamma_l, \gamma_{\text{CoT}})$ with $\gamma_{\text{CoT}} = \alpha\gamma_e + \beta\gamma_l$.

- Current step $c_i$: [step_ci]
- Parent trace $\mathbb{C}_i^{pa}$: [parent_trace]
- Background knowledge $\mathbb{K}_i$: [background_knowledge]
- Question $Q$: [question]
- Weights $(\alpha, \beta)$ with $\alpha, \beta \geq 0$ and $\alpha + \beta = 1$

---

*Task-grounded scorers (interface):*

- **Answer scorer** $S_{\mathrm{ans}} : \mathcal{Y} \to [0,1]$ — adequacy of the terminal output $c_n$ for $Q$ under the domain's correctness criteria.

- **Logical scorer** $S_{\log} : \mathcal{T}_{c_i} \times \mathcal{T}_{\mathbb{C}_i^{pa}} \to [0,1]$ — coherence of $(\mathbb{C}_i^{pa} \Rightarrow c_i)$ under domain rules.

**(1) Evidential causal effect** $\gamma_e \in [0,1]$

Conceptually contrast two rollouts: (a) the unmodified chain $(do(\emptyset))$, and (b) a chain where the influence of $c_i$ is removed $(do(c_i))$. Estimate the change in expected answer adequacy *using* $S_{\mathrm{ans}}$: report how much the presence of $c_i$ increases the expected $S_{\mathrm{ans}}(c_n)$ for $Q$ (background fixed). Output a scalar in $[0,1]$.
`[evidential_score]`

**(2) Logical causal effect** $\gamma_l \in [0,1]$

Contrast: (a) evaluating $c_i$ *with* parents provided $(do(\mathbb{C}_i^{pa}))$ vs. (b) evaluating $c_i$ *without* parents, i.e., $S_{\log}(c_i, \emptyset)$ under $do(\emptyset)$. Estimate the coherence gain *using* $S_{\log}$. Output a scalar in $[0,1]$.
`[logical_score]`

**(3) Combined CACE** $\gamma_{\mathrm{CoT}} \in [0,1]$

Combine the two effects using the given weights: `[causal_score]` where `[causal_score]` $= \alpha \cdot$ `[evidential_score]` $+ \beta \cdot$ `[logical_score]`.

**How formulas are embedded** We explicitly *name* and *type* the scorers in the prompt to bind judgments to measurable, domain-grounded quantities:

$$S_{\mathrm{ans}} : \mathcal{Y} \to [0,1], \qquad S_{\log} : \mathcal{T}_{c_i} \times \mathcal{T}_{\mathbb{C}_i^{pa}} \to [0,1].$$

When domain oracles exist (e.g., tests/CAS/rules), the evaluator queries them to obtain these values; otherwise, frozen, calibrated verifiers are used as surrogates. The requested `[evidential_score]` serves as a proxy for the intervention contrast

$$\mathbb{E}[S_{\mathrm{ans}}(c_n) \,|\, do(\emptyset),\, Q,\, \mathbb{K}_n] \quad \text{vs.} \quad \mathbb{E}[S_{\mathrm{ans}}(c_n) \,|\, do(c_i),\, Q,\, \mathbb{K}_n],$$

and `[logical_score]` is a proxy for

$$\mathbb{E}[S_{\log}(c_i, \mathbb{C}_i^{pa}) \,|\, do(\mathbb{C}_i^{pa}),\, Q,\, \mathbb{K}_i] \quad \text{vs.} \quad \mathbb{E}[S_{\log}(c_i, \emptyset) \,|\, do(\emptyset),\, Q,\, \mathbb{K}_i],$$

aligning with Eq. equation 2. The combined `[causal_score]` instantiates Eq. equation 3 via $\gamma_{\mathrm{CoT}} = \alpha\gamma_e + \beta\gamma_l$.

**Estimation protocol (stability).** Following Definition 1, we run $m$ independent evaluations with fixed wording and controlled stochasticity to obtain $\{(\gamma_e^{(r)}, \gamma_l^{(r)}, \gamma_{\mathrm{CoT}}^{(r)})\}_{r=1}^m$, compute Monte Carlo means $\widehat{\gamma}_e, \widehat{\gamma}_l$, and form $\widehat{\gamma}_{\mathrm{CoT}} = \alpha\,\widehat{\gamma}_e + \beta\,\widehat{\gamma}_l$. Bootstrap confidence intervals are used to (i) reduce sensitivity to decoding randomness and seed choice Xie et al. (2024b); Mora-Cross et al. (2024) and (ii) quantify the finite-sample variance of $\widehat{\gamma}_e$ and $\widehat{\gamma}_l$ or Various (2024). Each invocation must return three scalars in $[0,1]$ in the fixed order $(\gamma_e, \gamma_l, \gamma_{\mathrm{CoT}})$ (with $\gamma_{\mathrm{CoT}} = \alpha\gamma_e + \beta\gamma_l$); any out-of-range values are clipped to $[0,1]$. This preserves alignment with the intervention semantics in Eq. equation 2 and the aggregation in Eq. equation 3 while matching the notation used in the main text.

A.5.2 IMPLEMENTATION OF STEPWISE CAUSALITY UPDATING FUNCTION

The stepwise causality updating function $f_{\mathrm{update}}$ is implemented via a two-stage prompt-based procedure, as formally defined in Definition A.5.2. The objective is to revise a faulty reasoning step $\dot{c}_i$ by generating and selecting a corrected version $c_i'$ that maximizes its causal effectiveness within the reasoning chain. To operationalize $f_{\mathrm{update}}$, we decompose it into two prompt-embedded subroutines: (1) the *causalization module* $f_{\mathrm{cau}}$ generates a set of candidate revisions $\mathbf{c}_i = \{c_i^{(1)}, \ldots, c_i^{(k)}\}$; (2) the *refinement module* $f_{\mathrm{refine}}$ evaluates these candidates and selects the most effective one.

In the causalization stage, the LLM acts as a domain-specific reasoning agent (e.g., mathematician, physician, or analyst depending on the context $Q \cup \mathbb{K}_i$), producing diverse and causally plausible

candidates based on the faulty step $\dot{c}_i$ and its parent trace $\mathbb{C}_i^{pa}$. This generation process embeds the intent of improving both logical coherence (targeting $\gamma_l$) and evidential relevance (targeting $\gamma_e$), thus aligning with the underlying causal model. In the refinement stage, each candidate $c_i^{(j)}$ is re-evaluated using the same judgment mechanism $f_{\text{judge}}$ that computes the CoT causal score $\gamma_{\text{CoT}}$. The candidate with the highest score is returned as the final corrected step $c_i'$. This two-stage composition ensures that $f_{\text{update}} = f_{\text{refine}} \circ f_{\text{cau}}$ can be instantiated entirely through prompt engineering—without modifying model weights—thus making the step-level causal correction process both interpretable and modular within the LLM framework.

---

**Stepwise Causal Updating Prompt**

**Stage 1: Causalization Prompt (for $f_{\text{cau}}$)** You are revising a faulty reasoning step $\dot{c}_i$ in a step-by-step solution to the question `[Q]`. Use only the provided parent trace and background knowledge. Produce $k$ candidate corrections that explicitly target causal improvement. **Inputs**

- Faulty step $\dot{c}_i$: `[step_dot_ci]`
- Parent trace $\mathbb{C}_i^{pa}$: `[parent_trace]`
- Background knowledge $\mathbb{K}_i$: `[background_knowledge]`
- Question $Q$: `[question]`

**Requirements (all must hold)**

1. *Logical consistency:* Each candidate must be coherent with $\mathbb{C}_i^{pa}$ (no contradictions or unsupported leaps).

2. *Evidential relevance:* Each candidate must help address `[Q]` under $\mathbb{K}_i$ (no extraneous content).

3. *Semantic diversity:* The set $\{c_i^{(1)}, \ldots, c_i^{(k)}\}$ should contain meaningfully different revisions (not mere paraphrases).

**Output format**

- A numbered list of $k$ candidates: `[cand_1]`, `[cand_2]`, ..., `[cand_k]`.

---

**Stage 2: Refinement Prompt (for $f_{\text{refine}}$)**

Given candidates $\{c_i^{(1)}, \ldots, c_i^{(k)}\}$, select the one with the highest CoT Average Causal Effect (CACE) as defined in Eq. equation 3.

**Inputs**

- Candidates: `[cand_1]`, ..., `[cand_k]`
- Parent trace $\mathbb{C}_i^{pa}$: `[parent_trace]`
- Background knowledge $\mathbb{K}_i$: `[background_knowledge]`
- Question $Q$: `[question]`
- Weights $(\alpha, \beta)$ with $\alpha, \beta \geq 0$, $\alpha + \beta = 1$

**Scoring protocol**

1. For each candidate $c_i^{(j)}$, call the judgment function to obtain three scalars in $[0, 1]$:

$$(\gamma_e^{(j)}, \gamma_l^{(j)}, \gamma_{\text{CoT}}^{(j)}) = f_{\text{judge}}(c_i^{(j)}, \mathbb{C}_i^{pa}, Q, \mathbb{K}_i), \quad \gamma_{\text{CoT}}^{(j)} = \alpha\,\gamma_e^{(j)} + \beta\,\gamma_l^{(j)}.$$

2. Select $c_i' = \arg\max_j \gamma_{\text{CoT}}^{(j)}$.

**Output format**

- **Corrected step** $c_i'$: `[best_candidate_text]`
- **Scores for** $c_i'$: `[gamma_e_best]`, `[gamma_l_best]`, `[gamma_cot_best]`
- (Optional) Per-candidate scores: a brief table of $(\gamma_e^{(j)}, \gamma_l^{(j)}, \gamma_{\text{CoT}}^{(j)})$ for transparency.

**Language–formula alignment and stability.**  The refinement stage applies the same intervention semantics as Eq. equation 2 and the aggregation rule of Eq. equation 3 *to each candidate*. For a candidate $c_i^{(j)}$, we invoke the judgment mechanism (Def. A.5.1) with an identical prompt template and scorer instantiation across candidates, and repeat it $m$ times under controlled stochasticity to obtain samples $(\gamma_e^{(j,r)}, \gamma_l^{(j,r)}, \gamma_{\text{CoT}}^{(j,r)}) \in [0,1]^3$. We then compute per–candidate Monte Carlo estimates

$$\widehat{\gamma}_e^{(j)} = \tfrac{1}{m} \sum_{r=1}^m \gamma_e^{(j,r)}, \qquad \widehat{\gamma}_l^{(j)} = \tfrac{1}{m} \sum_{r=1}^m \gamma_l^{(j,r)}, \qquad \widehat{\gamma}_{\text{CoT}}^{(j)} = \alpha\, \widehat{\gamma}_e^{(j)} + \beta\, \widehat{\gamma}_l^{(j)},$$

with bootstrap confidence intervals. All outputs are clipped to $[0,1]$ and reported in the fixed order $(\gamma_e, \gamma_l, \gamma_{\text{CoT}})$ for comparability. The corrected step is $c_i' = \arg\max_j \widehat{\gamma}_{\text{CoT}}^{(j)}$ (ties broken by larger $\widehat{\gamma}_l^{(j)}$, then $\widehat{\gamma}_e^{(j)}$).

By *(i)* generating candidates that explicitly target logical and evidential improvements and *(ii)* selecting via scorer–grounded, intervention–based contrasts computed with a uniform protocol across candidates, this procedure reduces free–form arbitrariness, ties revisions to task semantics, and preserves consistency with the SCM–based objective throughout.

# B  SUPPLEMENTARY EXPLANATION TO CRBENCH

## B.1  CoT PROCESS ERROR DATASET FOR CAUSAL LABELING IN CRBENCH

We select the PROCESSBENCH (PB) dataset Zheng et al. (2024) for labeling in CRBench. It consists of 3,400 test cases, primarily focused on competition and Olympiad-level math problems. Each test case contains a step-by-step solution with error location annotated by human experts.

Specifically, it contains queries from the following four datasets:

**GSM8K** Cobbe et al. (2021) contains high quality linguistically diverse grade school math problems.

**Math** Hendrycks et al. (2021b) is a challenging competition math problems dataset. Each problem requires a complete step-by-step solution to arrive at the correct answer.

**OlympiadBench** He et al. (2024) is an Olympiad-level bilingual multimodal science benchmark that contains Olympiad-level math and physics competition problems, including the Chinese college entrance examination. Each problem requires expert-level annotations to complete step-by-step reasoning. We focus OlympiadBench's physics part in our experiment.

**Omni-MATH** Gao et al. (2024) is a mathematics-focused, comprehensive and challenging benchmark specifically designed to assess LLMs' mathematical reasoning ability at the Olympiad level. It is rigorously manually annotated. The quarys are carefully divided into more than 33 sub-areas covering more than 10 different difficulty levels.

## B.2  BASE DATASET FOR THE GENERATION OF CAUSALLY ERRONEOUS REASONING DATA IN CRBENCH

We generate new causally erroneous reasoning data based on a high-quality reasoning dataset distilled from DeepSeek-R1 Guo et al. (2025); DeepSeek-AI (2025). To generate the CRBench data, we introduce causal errors into the steps of CoT. The base dataset is primarily sourced from:

1. Bespoke-Stratos-17k Labs (2025) is a reasoning dataset consisting of questions, reasoning paths, and answers. It was created by replicating and improving the Berkeley Sky-T1 data pipeline using SFT distillation data from DeepSeek-R1.

2. Open-Thoughts-114k Team (2025), a synthetic reasoning dataset containing 114k high-quality examples, covering a diverse range of question types representative of domains where LLMs are widely applied.

3. OpenThoughts2-1M Team (2025) builds upon OpenThoughts-114k dataset, augmenting it with existing datasets like OpenR1, as well as additional math and code reasoning data. This dataset was used to train OpenThinker2-7B and OpenThinker2-32B Team (2025).

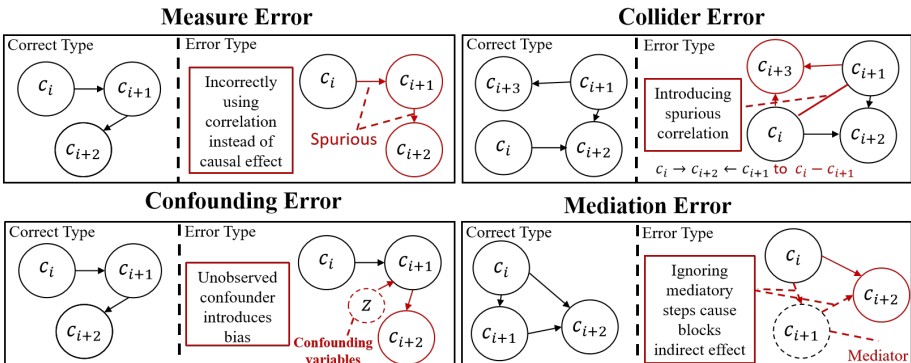

Figure 8: **Measure error Chwialkowski et al. (2014); Scheines & Ramsey (2017)**: Measure error refers to the incorrect use of correlation indicators instead of causal indicators when measuring causal relations between steps, or the use of inappropriate causal measures (like CACE) when estimating causal effects. **Collider error Schneider (2020); Holmberg & Andersen (2022)**: Collider error refers to the incorrect control or selection of a "collider" in CoT, which introduces false correlation. A collider is a steps that is affected by two unrelated steps at the same time. If this collider is incorrectly controlled during analysis, it will cause false correlations between originally unrelated steps. Due to selection bias when selecting samples, two originally unrelated steps appear to have a causal relation. **Confounding error Cinelli et al. (2019):** Confounding error refers to the omission of a confounder in CoT, leading to an observed causal effect that is not genuine but rather driven by a common influencing steps. It can also occur when steps that should not be included in the reasoning are considered, such as residual information from a previous quary, biases within the model, hallucinations, and other misleading factors. **Mediation error Pearl (2014):** Mediation error refers to the incorrect interpretation of the role of the mediating step in CoT, which may be due to incorrect control of the mediating step, incorrect addition of the mediating step, or ignoring mediating steps.

Given the large scale of both datasets, we use LLMs to filter and curate the most challenging examples—those deemed difficult by the models—across various question types. Through careful manual controls, CRBench ultimately includes the following categories of queries:

**Code generation queries**: 1.TACO Li et al. (2023d) is a benchmark for code generation with 26443 problems. It can be used to evaluate the ability of language models to generate code from natural language specifications.

2.APPS Hendrycks et al. (2021a) is a benchmark for code generation with 10000 problems. It can be used to evaluate the ability of language models to generate code from natural language specifications.

3.CodeContests Li et al. (2022) is a competitive programming dataset for machine-learning. This dataset was used when training AlphaCode Li et al. (2022). Problems include test cases in the form of paired inputs and outputs, as well as both correct and incorrect human solutions in a variety of languages.

4.LiveCodeBench Jain et al. (2024) is a "live" updating benchmark for holistically evaluating code related capabilities of LLMs. Particularly, it evaluates LLMs across a range of capabilties including code generation, self-repair, test output prediction, and code execution. This is the code generation scenario of LiveCodeBench. It is also used for evaluating self-repair using test case feedback.

**Mathematical reasoning queries**: 1.NuminaMath LI et al. (2024) includes approximately 860k math problems, where each solution is formatted in a CoT manner. The sources of the dataset range from Chinese high school math exercises to US and international mathematics olympiad competition problems. The data were primarily collected from online exam paper PDFs and mathematics discussion forums.

2.AIME 2024 dataset contains problems from the American Invitational Mathematics Examination (AIME) 2024. AIME is a prestigious high school mathematics competition known for its challenging mathematical problems.

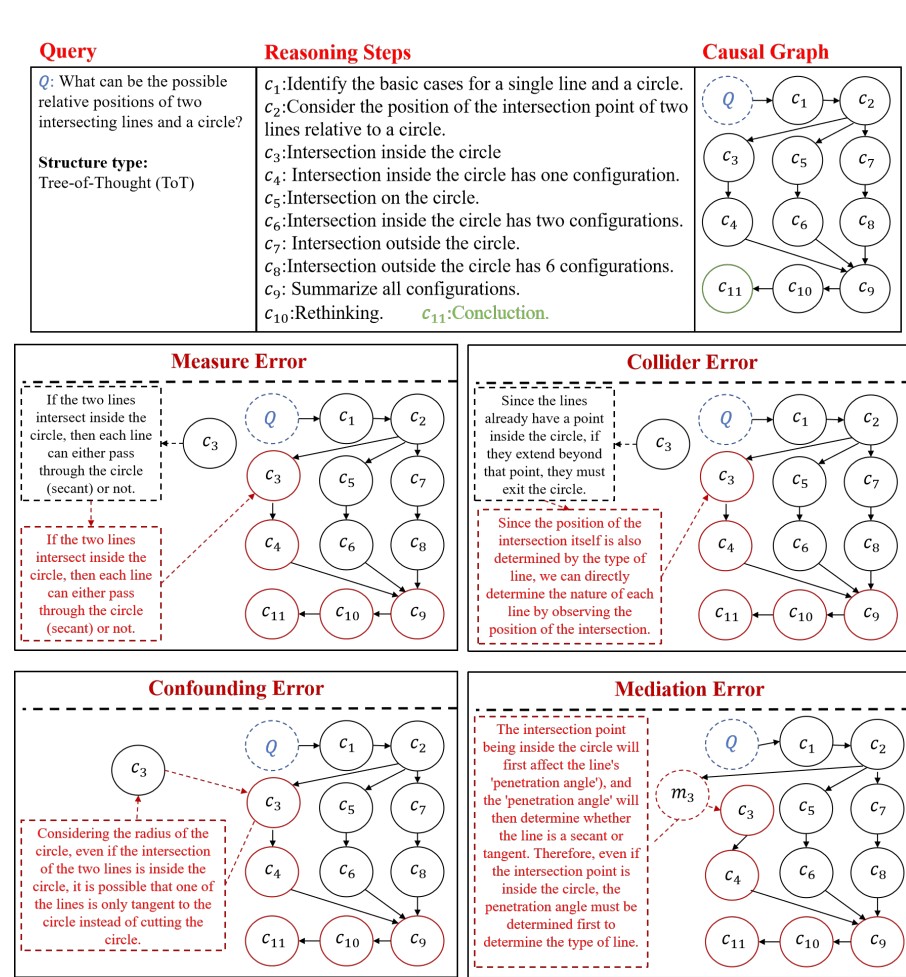

Figure 9: The example of generated causally erroneous reasoning data in CRBench. **Causality measure error:** In the process of determining that "when the intersection is inside the circle, each line must be a secant," the reasoning mistakenly overstates the impact of the intersection point's location. It erroneously asserts that "as long as the intersection is inside the circle, each line must intersect the circle at two points," thereby ignoring the possibility that a line might only intersect the circle at one point (which would be a tangent),leading to a causality measure error. **Collider error:** When considering the impact of the intersection point's position on the relation between the lines and the circle, the reasoning mistakenly treats the intersection position (inside, on, outside) as a "collider" that is simultaneously determined by both the type of the lines and the circle's position. This error mixes independent factors. **Confounding Error:** In the reasoning, an unrelated external factor is incorrectly introduced as a confounding step. It is mistakenly assumed that this step affects both the position of the intersection and the number of intersection points between the lines and the circle, which leads to an incorrect derivation of the number of possible configurations.This incorrectly introduces the circle's radius as a confounder, mixing up the originally clear causal relation based solely on the intersection point's location, hence causing a confounding error. **Mediation error:** Here, an unneeded and non-existent mediator step called 'penetration angle' is introduced, thereby misrepresenting the causal relation between the intersection location and the line type, resulting in a mediation error, mistakenly assuming that the causal relation between the intersection point's location and the line type is transmitted through this mediator, which then leads to a misinterpretation of the relations among variables.

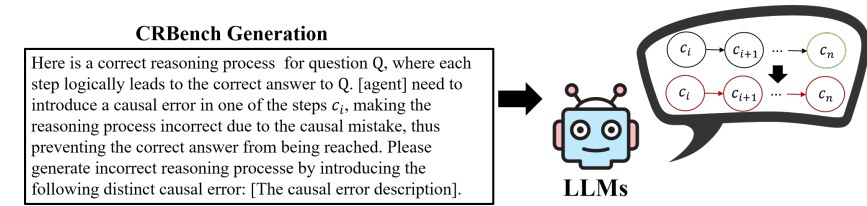

Figure 10: A formal description of how to generate CRBench data using LLMs. Given that the step introducing a causal error is $c_i \in \mathbb{C}$, through a standardized prompt, LLMs induce a corresponding causal error that disrupts the causal connection with $\mathbb{C}_i^{pa}$, thereby causing subsequent reasoning to lack correct causal relations and ultimately leading to incorrect reasoning outcomes.

3.MATH-500 contains a subset of 500 problems from the MATH benchmark that OpenAI created in their "Let's Verify Step by Step paper" Lightman et al. (2023). It based on RM800K which is a process supervision dataset containing 800,000 step-level correctness labels for model-generated solutions to problems.

**Scientific QA queries**: 1. Chemistry dataset Li et al. (2023b) is composed of 20K problem-solution pairs obtained using GPT-4. The dataset problem-solutions pairs generating from 25 chemistry topics, 25 subtopics for each topic and 32 problems for each "topic,subtopic" pairs.

2.Biology dataset Li et al. (2023b) is composed of 20K problem-solution pairs obtained using GPT-4. The dataset problem-solutions pairs generating from 25 biology topics, 25 subtopics for each topic and 32 problems for each "topic,subtopic" pairs.

3.Physics dataset Li et al. (2023c) is composed of 20K problem-solution pairs obtained using GPT-4. The dataset problem-solutions pairs generating from 25 physics topics, 25 subtopics for each topic and 32 problems for each "topic,subtopic" pairs.

**Puzzle-solving queries**: 1.RiddleSense Lin et al. (2021) is a multiple-choice question answering dataset consisting of 5.7k riddle-style commonsense questions. It is designed to evaluate complex reasoning abilities such as figurative language understanding, counterfactual reasoning, and higher-order commonsense. As the first large-scale dataset of its kind, RiddleSense reveals a significant performance gap between state-of-the-art models and humans, highlighting the need for further research in advanced natural language understanding and linguistic creativity.

### B.3 CAUSAL ERRORS COMMONLY FOUND IN COT REASONING

As shown in Figure 8, four types of causal errors commonly found in CoT reasoning are defined and formally illustrated. These errors can lead to the formation of incorrect causal between steps and incorrect reasoning steps, ultimately resulting in incorrect reasoning answers.

### B.4 IMPLEMENTATION OF CRBENCH GENERATION

We employ a unified prompt that introduces causal errors while preserving reasoning coherence. The causal errors types are referred to as [Causal error description], and an example of the generating process is illustrated in Figure 10. Data generation of CRBench is performed using the R1Distill-Qwen-72B model DeepSeek-AI (2025). (Similarly, the prompts shown in the figure are merely illustrative; any reasonable prompt design tailored to different CoT reasoning scenarios is valid and feasible.)

We use CoT in Fig 7 as an example to show how CRBench introduce four types of causal errors to high-quality CoT in Fig 9 by the generating process in 10.

# C  Additional Experiments and Discussions on CauCoT

## C.1  Detailed Description of How to Implement Intervention

This section elaborates the how to finish $do(c_i)$ in Experiment. We describe two practical protocols, their motivation, and diagnostics. Throughout, effects are evaluated with $S_{\text{ans}}(\cdot)$ and $S_{\text{log}}(\cdot)$ as in Eq. 2–3.

**(P1) Ablate.**  We keep $(Q, K, \mathbb{C}_i^{pa})$ unchanged and delete the textual content of $c_i$ before decoding the remaining steps and the final answer $a$. Decoding hyperparameters are identical to the original run. This realizes an intervention in which the information carried by $c_i$ is removed while preserving the parental context. In practice, we instantiate deletion by omitting $c_i$ from the visible trace passed to the model.

**(P2) Counterfactual re-generation.**  We keep $(Q, K, \mathbb{C}_i^{pa})$ unchanged and re-generate the textual content of $c_i$ with the same decoding hyperparameters as the original run; we then continue decoding the remaining steps and the final answer. This realizes a counterfactual variation of $c_i$ conditioned on the same parental context, probing outcome sensitivity to alternative but plausible step content.

**Why these protocols are reasonable.**  Both (P1) and (P2) conform to the common intervention reading of modifying a node while holding non-descendants fixed Pearl (2009). In CoT/ToT-style stepwise reasoning Wei et al. (2022); Yao et al. (2023), $(Q, K, \mathbb{C}_i^{pa})$ plays the role of observed parents; (P1) simulates removal of the contribution of $c_i$, whereas (P2) simulates a counterfactual variant of $c_i$ under the same parents. These are standard operationalizations of interventions when variables are textual and the generator is a probabilistic decoder.

**Limitations.**  Textual interventions approximate but do not equal ideal interventions on semantic variables; deletion may remove formatting cues helpful for decoding, and re-generation may shift discourse style. We mitigate these issues by holding $(Q, K, \mathbb{C}_i^{pa})$ fixed, keeping decoding settings unchanged, and averaging over re-samples.

## C.2  Detailed Description of Faithfulness Evaluation

In addition to accuracy, we evaluate the Faithfulness (Faith) of CoT outputs, which captures the consistency between the final answer and the CoT. Faithfulness is rated on a 1–5 scale, where 1 denotes minimal consistency and 5 indicates high alignment.

**Causalized, step-weighted metric (C-Faith).**  We ground faithfulness in the causal structure used by CauCoT. Let $\mathbb{C} = \{c_1, \ldots, c_n\}$ be the CoT and $\gamma_{\text{CoT}}(c_i, Q, \mathbb{K})$ the stepwise causal contribution defined in the main text. We assign a nonnegative weight to each step and aggregate local coherence:

$$w_i \;=\; \frac{\max\{\gamma_{\text{CoT}}(c_i, Q, \mathbb{K}),\, 0\}}{\sum_{j=1}^{n-1} \max\{\gamma_{\text{CoT}}(c_j, Q, \mathbb{K}),\, 0\}}, \qquad \text{C-Faith}(\mathbb{C}) \;=\; \sum_{i=1}^{n-1} w_i \cdot S_{\text{log}}(c_i, \mathbb{C}_i^{pa}),$$

where $S_{\text{log}} \in [0, 1]$ is the step-level coherence map defined in the method section. This causally-weighted average discounts decorational or incoherent steps (low $\gamma_{\text{CoT}}$) and emphasizes steps that are both locally coherent and causally contributive. To report on the same 1–5 scale as the main metric, we linearly rescale:

$$\text{Faith}_{\text{causal}} \;=\; 1 + 4 \cdot \text{C-Faith}(\mathbb{C}) \;\; \in [1, 5].$$

*Diagnostics (not part of the score).* For sanity checks, we compute the counterfactual answer drop for step $c_i$, $\Delta\text{Ans}(i) = \mathbb{E}[S_{\text{ans}}(c_n) \mid do(\emptyset)] - \mathbb{E}[S_{\text{ans}}(c_n) \mid do(c_i)]$, and report its rank correlation with $\gamma_e(c_i)$; strong alignment indicates that steps deemed evidentially causal are also those whose removal most degrades answer adequacy.

**Human evaluation with a unified 1–5 rubric (H-Faith).**  Each instance is rated independently by $\geq 3$ raters with advanced CS/ML background (familiar with LLMs and their evaluation; at least a Master's degree); for domain-specific subsets (e.g., math), at least one domain-qualified rater is included. Raters are *double-blinded* to model identity and condition; item order is randomized. Before

annotation, raters must (i) pass a 10-item qualification quiz (gold answers; $\geq 80\%$ required) and (ii) complete a 6-item calibration pack (one anchor per score 1–5 plus a borderline case) with feedback. During annotation, we interleave 10% gold items and 5% attention checks; raters failing either are excluded and their items re-assigned. Raters use a 1–5 scale based on three guiding questions: (1) Are there any logical leaps? (2) Does the reasoning contain factual errors? (3) Does the conclusion truly follow from the reasoning? We provide anchor definitions to standardize the scale:

- **1** (Minimal): major logical leaps or contradictions; factual errors; conclusion does not follow.
- **2** (Low): multiple issues; partial support at best; notable gaps undermine the conclusion.
- **3** (Moderate): generally coherent with minor lapses; limited but present support for the conclusion.
- **4** (High): coherent and mostly accurate; conclusion follows with small caveats.
- **5** (Maximal): fully coherent and accurate; conclusion is clearly and directly supported by the steps.

**Final Faith score and reporting.**    Our final Faith score combines the two parts on the common 1–5 scale:
$$\text{Faith} \;=\; \eta \cdot \text{Faith}_{\text{causal}} \;+\; (1 - \eta) \cdot \text{Faith}_{\text{human}}, \qquad \eta \in [0, 1],$$
with $\eta = 0.5$ by default. To ensure commensurability, we verify that both components match the anchor distribution on a small development split; if minor drift is detected, we apply a monotone (isotonic) recalibration to map each component to the anchor-consistent 1–5 range without changing item ordering. We report means with bootstrap 95% CIs, and conduct paired permutation tests (item-wise) with Holm–Bonferroni correction. As a robustness check, we repeat all comparisons across random seeds/temperatures and report variability bands; where applicable, we fit a linear mixed-effects model (method as fixed effect; item and rater as random intercepts) to confirm significance under rater/item heterogeneity.

### C.3    EXAMPLES OF CAUCOT CORRECTS CAUSAL ERRORS IN CRBENCH.

The following section presents examples of how CauCoT corrects causal error in generated data from the CRBench. Each entry is indexed by an "id". The "problem" field provides the corresponding query. The "ground truth solution" contains the original, correct reasoning chain. The "CRBench CoT" column introduces injected causal errors into the original reasoning, with erroneous steps highlighted in red. The "CauCoT CoT" displays the corrected version produced by CauCoT, where the corrected steps are marked in green. While not incorrect, the parts that underwent further improvement are highlighted in orange.

---

**An Example of the Correction for Confounding Error by CauCoT**

**ID:** 18

**Problem:** In a triangle, two sides are given. For what value of the angle between them will the area of the triangle be the greatest?

**Ground Truth Solution:**

- 1. Given a triangle with two sides, let's denote these sides as $(a)$ and $(b)$. The angle between these two sides is denoted as $(\theta)$.
- 2. The area $(A)$ of a triangle when two sides and the included angle are known can be calculated using the formula: $A = \frac{1}{2}ab\sin\theta$.
- 3. Since the sides $(a)$ and $(b)$ are constant, the area $(A)$ will be maximized when $\sin\theta$ is maximized.
- 4. The sine function $\sin\theta$ reaches its maximum value of 1 when $\theta = 90°$ (or $\frac{\pi}{2}$ radians), because $\sin 90° = 1$.
- 5. Therefore, the area of the triangle will be the largest when the angle between the sides is a right angle, i.e., $\theta = 90°$.

---

- **Conclusion:** The area of the triangle is maximized when the angle is $\boxed{90°}$.

**CRBench CoT (Erroneous Reasoning):**

- 1. Given a triangle with two sides, let's denote these sides as $(a)$ and $(b)$. The angle between these two sides is denoted as $(\theta)$.

- 2. The area $(A)$ of a triangle when two sides and the included angle are known can be calculated using the formula: $A = \frac{1}{2}ab\sin\theta$.

- 3. Since the sides $(a)$ and $(b)$ are constant, the area $(A)$ will be maximized when $\sin\theta$ is maximized.

- 4. The sine function $\sin\theta$ reaches its maximum value of 1 when $\theta = 180°$ (or $\pi$ radians), because $\sin 180° = 1$.

- 5. Therefore, the area of the triangle will be the largest when the angle is $180°$.

- Conclusion: The area of the triangle is maximized when the angle between the given sides is $\boxed{180°}$.

**Error Type:** Confounding Error

**CauCoT CoT (Corrected Reasoning):**

- 1. Given a triangle with two sides, let's denote these sides as $(a)$ and $(b)$. The angle between these two sides is denoted as $(\theta)$.

- 2. The area $(A)$ of a triangle when two sides and the included angle are known can be calculated using the formula: $A = \frac{1}{2}ab\sin\theta$.

- 3. Since the sides $(a)$ and $(b)$ are constant, the area $(A)$ will be maximized when $\sin\theta$ is maximized.

- 4. The sine function $\sin\theta$ reaches its maximum value of 1 when $\theta = 90°$ (or $\frac{\pi}{2}$ radians), as this is the point where the sine curve peaks.

- 5. Therefore, the area of the triangle will be the largest when the angle between the sides is $90°$ (or $\frac{\pi}{2}$ radians).

- Conclusion: The area of the triangle is maximized when the angle between the given sides is $90°$. The sine function reaches its maximum value at $90°$.

---

**An Example of the Correction for Measure Error by CauCoT**

**ID:** 260

**Problem:** Each side of square $ABCD$ is 8. A circle is drawn through $A$ and $D$ so that it is tangent to $BC$. What is the radius of this circle?
(A) 4    (B) 5    (C) 6    (D) $4\sqrt{2}$    (E) 5.25

**Ground Truth Solution:**

- To solve for the radius $r$ of the circle that passes through vertices $A$ and $D$ of square $ABCD$ and is tangent to side $BC$, we follow these steps:

- 1. Identify key points and geometry:
    - Each side of the square $ABCD$ is of length 8.
    - Let the radius of the circle be $r$ and the center be $O$.
    - Since the circle passes through $A$ and $D$, and is tangent to $BC$, the distance from $O$ to $BC$ is $r$.

- 2. Set up coordinates:
    - Let $A = (0, 8), B = (8, 8), C = (8, 0), D = (0, 0)$, and $O = (h, k)$.
    - Since the circle is tangent to $BC$, it implies $k = r$.

- 3. Use geometry of the circle:
  - The midpoint of $AD$ is $(0, 4)$, and the center $O$ lies on the perpendicular bisector of $AD$.
- 4. Calculate distance relations:
  - Length $AP = \frac{8}{2} = 4$.
  - In right triangle $OAP$: $OA = r$, and $OP = 8 - r$.
- 5. Apply the Pythagorean theorem:
  - $r^2 = AP^2 + OP^2 = 4^2 + (8-r)^2 = 16 + 64 - 16r + r^2$.
  - Subtract $r^2$: $0 = 80 - 16r \Rightarrow r = 5$.
- **Conclusion:** The radius of the circle is $\boxed{5}$.

**CRBench CoT (Erroneous Reasoning):**

- 1. Same initial setup as ground truth (geometry and coordinates).
- 2. Coordinates and midpoint of $AD$ correctly identified as $(0, 4)$.
- 3.
  - OP was computed as $4 - r$ instead of $r - 4$ or $8 - r$, leading to:
  - $r^2 = 4^2 + (4-r)^2 = 16 + 16 - 8r + r^2$.
  - $0 = 32 - 8r \Rightarrow r = 4$.
- Conclusion: The reasoning leads to $\boxed{4}$.

**Error Type:** Measure Error

**CauCoT CoT (Corrected Reasoning):**

- 1. Geometry and coordinates are consistent with the ground truth.
- 2. The center $O = (h, r)$, midpoint of $AD$ is $(0, 4)$, and lies on the perpendicular bisector.
- 3. In triangle $OAP$:
  - Length $AP = 4$, and $OP = r - 4$.
  - Pythagorean theorem: $r^2 = 4^2 + (r-4)^2$.
  - Expand: $r^2 = 16 + r^2 - 8r + 16$, simplify to $0 = -8r + 32 \Rightarrow r = 5$.
- **Conclusion:** The radius of the circle is $\boxed{5}$.

---

### An Example of the Correction for Mediation Error by CauCoT

**ID:** 30

**Problem:** The New Year's Eve performance includes four acts: singing, dancing, acrobatics, and skits. If the requirements are that singing is not in the 4th position, dancing is not in the 3rd position, acrobatics are not in the 2nd position, and skits are not in the 1st position, then how many different arrangements of the program are there that satisfy the above requirements?

**Ground Truth Solution:**

- We are to count the number of valid permutations of {Singing (S), Dancing (D), Acrobatics (A), Comedy (C)} with the following position constraints:
  - S *not* in position 4;
  - D *not* in position 3;
  - A *not* in position 2;
  - C *not* in position 1.
- 1. For position 1, C is not allowed, so valid choices are {S, D, A} — 3 options.

- 2. For position 2, A is excluded. Among the remaining 3 acts (after fixing position 1), exclude A to get 2 valid options.

- 3. For position 3, D is excluded. After fixing positions 1 and 2, among the 2 remaining acts, exclude D to get 1 valid choice.

- 4. For position 4, S is excluded. The last remaining act must not be S — which is ensured by the above setup.

- Thus, the total valid permutations: $3 \times 2 \times 1 \times 1 = 6$.

- However, enumerating all permutations that satisfy all constraints gives 9 valid sequences, indicating the above exclusion logic undercounts due to hidden mediators (i.e., dependencies among steps).

- **Conclusion:** The correct number of valid permutations is $\boxed{9}$.

**CRBench CoT (Erroneous Reasoning):**

- 1. The CoT correctly identifies the acts and position constraints.

- 2. The reasoning applies stepwise filtering:

  - Position 1: exclude C $\Rightarrow$ 3 choices;
  - Position 2: exclude A $\Rightarrow$ 2 choices;
  - Position 3: exclude D $\Rightarrow$ 1 choice;
  - Position 4: exclude S $\Rightarrow$ 1 choice.

- 3. Based on this independence assumption, it computes $3 \times 2 \times 1 \times 1 = 6$ permutations.

- The stepwise decomposition implicitly assumes that constraints apply independently, failing to consider joint effects.

- Conclusion: The count is $\boxed{6}$.

**Error Type:** Mediation Error

**CauCoT CoT (Corrected Reasoning):**

- "1. **List the acts and positions.** - Acts: Singing (S), Dancing (D), Acrobatics (A), Comedy (C) - Positions: 1, 2, 3, and 4", "For position 1, since Skits cannot be placed here, we have three choices: Singing, Dancing, or Acrobatics.",

  - - Position 1: Comedy (C) cannot be in this position. Therefore, the valid choices for Position 1 are Singing (S), Dancing (D), and Acrobatics (A). So, there are 3 options for Position 1. - Position 2: After placing an act in Position 1, 3 acts are left. Acrobatics (A) cannot be in this position. Hence, the valid choices for Position 2 are the remaining two acts (not in Position 1) except Acrobatics. Therefore, there are 2 options for Position 2.- Position 3: After placing acts in Position 1 and Position 2, 2 acts are left. Dancing (D) cannot be in this position. Therefore, there is only 1 option for Position 3.- Position 4: After placing acts in Position 1, 2, and 3, there is only 1 act left, and Singing (S) cannot be in this position. Therefore, there is only 1 option for Position 4."

  - {For each valid choice of the first act, there are 2 valid choices for the second act, and the remaining two acts must be placed in the last two positions, giving us } 3 times 2 times 1 times 1 = 6.]

- 2.

  - Check each of the $4! = 24$ permutations.
  - Retain those that meet all four constraints jointly.

- 3. The valid permutations are:

  - (S, D, C, A), (S, A, C, D), (S, A, D, C), ...
  - (D, S, A, C), (D, A, S, C), ...
  - (A, S, D, C), (A, D, S, C), ...

  (Total: 9 valid permutations)

- **Conclusion:** The answer is $\boxed{9}$, obtained by evaluating the joint mediation effects rather than applying constraints in isolation.

---

### An Example of the Correction for Collider Error by CauCoT

**ID:** 58

**Problem:** Vasya wants to place numbers from 1 to 6 (each number used exactly once) into squares such that the following condition is met: if two squares are connected, the number in the higher square is greater. How many ways can this be done?

**Ground Truth Solution:**

- 1. **Understand the Structure:** The image (not shown here) has squares connected with arrows, representing causal constraints: a number in an upper square must be greater than the number in the connected lower square.

- 2. **Color Coding:** Color the squares based on their vertical position—upper (green) and lower (blue). The rule implies: number in green square ¿ connected blue square.

- 3. **Choose Numbers for Green Squares:** Choose any 2 out of 6 numbers for the green squares. Number of ways: $\binom{6}{2} = 15$.

- 4. **Determine Arrangement:**
    - The 2 chosen numbers for green squares must be placed in descending order (top to bottom).
    - The remaining 4 numbers are automatically assigned to blue squares in descending order (as per connection constraints).
    - Therefore, for each selection of green numbers, there is **exactly one** valid total arrangement.

- 5. **Final Count:** $15 \times 1 = 15$

- **Conclusion:** The total number of valid arrangements is $\boxed{15}$.

**CRBench CoT (Erroneous Reasoning):**

- Classifies squares and splits 6 numbers into 2 for green and 4 for blue squares.

- Computes $\binom{6}{2} = 15$ ways to choose numbers for green squares.

- It assumes that the 4 remaining numbers for blue squares can be freely arranged, and counts $4! = 24$ permutations, multiplying total to $15 \times 24 = 360$.

- This overcounts by violating the implicit causal constraint that blue squares must also follow a specific (descending) order; treating blue square arrangement as independent introduces collider bias.

- Total arrangements are $\boxed{360}$, which violates the causal structure of the ordering rules.

**Error Type:** Collider Error

**CauCoT CoT (Causal Correction):**

- Choose 2 of 6 numbers for green squares (top-level), rest go to blue squares.

- **Key Causal Insight:** Since both green and blue squares are subject to strict ordering (descending), only one valid arrangement exists per selection.

- Rejects the collider error in CRBench by enforcing that blue square arrangements are not independent choices—they are constrained by causal paths.

- Final Computation: $\binom{6}{2} \times 1 = 15$

- The count of valid arrangements is $\boxed{15}$, consistent with the ground truth.

Table 3: Correctness and faithfulness on PB datasets Zheng et al. (2024). The first column lists the dataset, the second the backbone LLMs. Columns 3–9 present zero-shot baselines (ZR, CoT, SC-CoT, ToT), and the last two columns report CauCoT. The metric row clarifies units: Acc is accuracy (%), Faith is on a 1–5 scale. We highlight the top three *zero-shot accuracy* results per row (among CoT/SC-CoT/ToT/CauCoT): red for 1st place, blue for 2nd place, orange for 3rd place.

| Dataset | Model | ZR | CoT | | SC-CoT | | ToT | | CauCoT | |
|---|---|---|---|---|---|---|---|---|---|---|
| | | Acc% | Acc% | Faith | Acc% | Faith | Acc% | Faith | Acc% | Faith |
| GSM8K | Qwen2.5-3b-Inst | 49.3 | 79.1 | 3.0 | 82.6 | 3.2 | 84.2 | 3.3 | 85.4 | 3.6 |
| | Qwen2.5-7b-Inst | 52.1 | 85.2 | 3.3 | 87.1 | 3.5 | 90.3 | 3.6 | 91.0 | 4.0 |
| | Llama-3-8B | 50.2 | 85.1 | 3.2 | 87.0 | 3.4 | 93.2 | 3.5 | 92.4 | 3.8 |
| | Qwen2.5-72b-Inst | 79.4 | 92.0 | 3.6 | 91.5 | 3.8 | 95.1 | 3.9 | 95.3 | 4.2 |
| | R1Distill-Qwen-32B | 85.2 | 95.0 | 4.2 | 96.1 | 4.3 | 97.0 | 4.4 | 97.2 | 4.5 |
| Math | Qwen2.5-3b-Inst | 43.5 | 37.2 | 2.4 | 40.4 | 2.6 | 45.8 | 2.8 | 64.3 | 3.5 |
| | Qwen2.5-7b-Inst | 51.1 | 38.2 | 2.6 | 42.3 | 2.8 | 48.4 | 3.0 | 68.2 | 3.7 |
| | Llama-3-8B | 44.2 | 52.0 | 3.0 | 56.2 | 3.2 | 60.5 | 3.4 | 65.4 | 3.9 |
| | Qwen2.5-72b-Inst | 57.3 | 82.0 | 3.8 | 85.1 | 4.0 | 86.2 | 4.1 | 88.0 | 4.3 |
| | R1Distill-Qwen-32B | 62.4 | 94.0 | 4.2 | 95.0 | 4.3 | 96.0 | 4.4 | 97.0 | 4.5 |
| OlympiadBench | Qwen2.5-3b-Inst | 6.2 | 11.0 | 1.8 | 13.1 | 2.0 | 15.3 | 2.2 | 39.0 | 3.2 |
| | Qwen2.5-7b-Inst | 8.0 | 15.0 | 2.0 | 18.2 | 2.3 | 22.1 | 2.5 | 51.0 | 3.6 |
| | Llama-3-8B | 7.0 | 13.0 | 2.0 | 16.1 | 2.2 | 19.2 | 2.4 | 44.0 | 3.4 |
| | Qwen2.5-72b-Inst | 12.0 | 20.0 | 2.5 | 24.0 | 2.8 | 30.4 | 3.1 | 63.0 | 3.9 |
| | R1Distill-Qwen-32B | 18.0 | 45.0 | 3.5 | 52.0 | 3.8 | 58.0 | 4.0 | 67.0 | 4.3 |
| OmniMath | Qwen2.5-3b-Inst | 14.0 | 17.0 | 2.2 | 20.3 | 2.4 | 24.7 | 2.6 | 45.0 | 3.4 |
| | Qwen2.5-7b-Inst | 18.1 | 24.0 | 2.5 | 28.2 | 2.7 | 32.3 | 2.9 | 56.0 | 3.7 |
| | Llama-3-8B | 16.0 | 25.0 | 2.6 | 29.2 | 2.8 | 34.1 | 3.0 | 44.0 | 3.6 |
| | Qwen2.5-72b-Inst | 22.0 | 36.0 | 3.2 | 41.2 | 3.5 | 48.3 | 3.7 | 68.0 | 4.1 |
| | R1Distill-Qwen-32B | 31.0 | 48.0 | 3.8 | 54.1 | 4.0 | 60.2 | 4.2 | 72.0 | 4.4 |

## C.4 EXPERIMENTS ON REAL CoT PROCESS ERROR DATA

Similar to the main text, we apply the CauCoT algorithm to judge and correct reasoning steps in CoT reasoning error: the PB dataset Zheng et al. (2024), evaluating correction effectiveness through reasoning performance on open-source LLMs. In the Appendix C.5.3, we provide examples of how CauCoT corrects causal errors in CoT from labeled CoT process error data in CRBench. Given the low computational demand of the PB dataset, we additionally perform hyperparameter and ablation studies to comprehensively validate the performance of CauCoT.

### C.4.1 IMPROVEMENTS IN THE CORRECTNESS OF CoT REASONING

Due to the relative simplicity of the questions in the PB dataset, LLMs with similar parameter scales exhibit only minor performance differences. Therefore, we present results using a representative subset of LLMs in our experiments. As shown in Table 3, CauCoT outperforms all other methods across all datasets and open-source large models. Notably, on more complex logical problem datasets, such as OlympiadBench and Omni-MATH, the improvement with CauCoT is more pronounced compared to relatively simpler datasets like GSM8K and Math. CauCoT significantly improves the accuracy of CoT reasoning and successfully corrects nearly all error steps in the PB dataset. The correctness improved by CauCoT in reasoning are substantial.

### C.4.2 IMPROVEMENTS IN THE CAUSALITY OF CoT REASONING

For the evaluation to the causality of CoT, we compare CauCoT with CoT and PB by analyzing the changes in causal effects between each step. Across four datasets, we report the heterogeneous effect (HE). The heterogeneous effect is defined as: $\text{HE} = \sqrt{n^{-1} \sum_{i=1}^{n} (c_n - (c_n \mid do(c_i)))^2}$. A higher HE means a step has a higher causal relation, higher causality.

Table 4: Understandability on PB datasets Zheng et al. (2024). The first column lists the dataset, the second the backbone LLMs. Columns 3–5 report *Accuracy% — HE* for CoT, PB, and CauCoT. HE is the heterogeneous effect $HE = \sqrt{n^{-1} \sum_{i=1}^{n} (c_n - (c_n \mid do(c_i)))^2}$; higher is more step-level causal influence (thus higher understandability).

| Dataset | Model | CoT (Acc% — HE) | PB (Acc% — HE) | CauCoT (Acc% — HE) |
|---|---|---|---|---|
| GSM8K | Qwen2.5-72B-Inst | 92 — 0.40 | 41 — 0.22 | **95 — 0.45** |
| | R1Distill-Qwen-32B | 95 — 0.36 | 52 — 0.28 | **98 — 0.42** |
| Math | Qwen2.5-72B-Inst | 82 — 0.32 | 45 — 0.24 | **88 — 0.34** |
| | R1Distill-Qwen-32B | 94 — 0.42 | 52 — 0.25 | **97 — 0.46** |
| OlympiadBench | Qwen2.5-72B-Inst | 20 — 0.16 | 17 — 0.15 | **63 — 0.45** |
| | R1Distill-Qwen-32B | 45 — 0.42 | 37 — 0.32 | **67 — 0.62** |
| OmniMath | Qwen2.5-72B-Inst | 36 — 0.30 | 30 — 0.14 | **68 — 0.62** |
| | R1Distill-Qwen-32B | 48 — 0.34 | 36 — 0.30 | **72 — 0.48** |

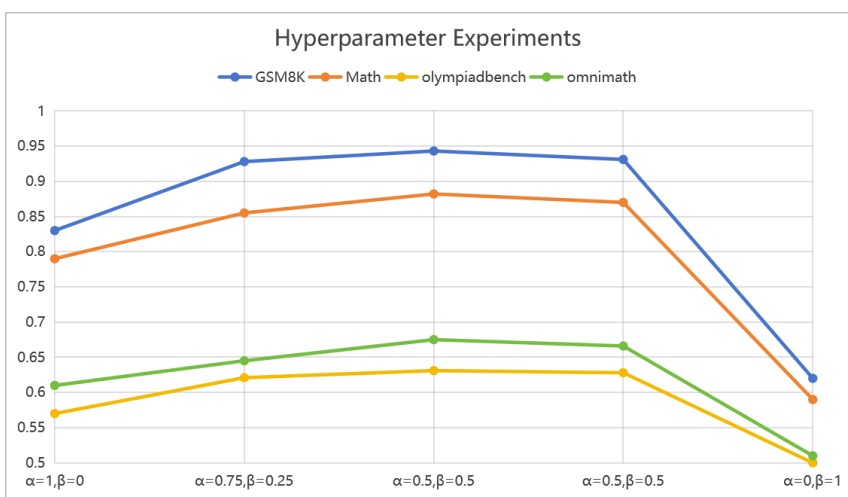

Figure 11: Experiments for $\alpha$ and $\beta$.

## C.5 Hyperparamter Experiments

### C.5.1 Experiments for $\alpha$ and $\beta$

Here, we discuss the setting of hyperparameters $\alpha$ and $\beta$ based on Qwen2.5-72b. As shown in the Figure 11, when both factors are balanced ($\alpha = \beta$), CauCoT achieves the strongest causality. This demonstrates that, to improve the performance of reasoning, **the correctness and understandability of CoT are equally important.**

### C.5.2 Experiments for $\sigma$

we analyze the setting of the $\sigma$ on Qianwen2.5-72b.

As shown in the Table 5, when the dataset is relatively complex, a higher $\sigma$ value makes updates more difficult to complete. This also highlights the necessity of setting $\sigma$ appropriately, allowing CauCoT to adjust the settings according to different scenarios to ensure feasibility.

### C.5.3 Examples of CauCoT Correct Real Reasoning Error

The following section presents examples of how CauCoT corrects causal errors in labeled data from CRBench. Each entry is indexed by an "id". The "problem" field provides the corresponding quary. The "CRBench CoT" column displays the reasoning error in steps that labeled as causal are marked

Table 5: $\sigma$ Evaluation. The first column lists the datasets used for evaluation. The second column shows the values of $\sigma$ set in the experiments, and the last column represents the proportion of CoT that is successfully achieved relaxed causal correctness.

| Dataset | $\sigma$ values | Relaxed causal correctness % |
|---|---|---|
| GSM8K | 0.5 | 100% |
| | 0.75 | 100% |
| | 0.9 | 100% |
| Math | 0.5 | 100% |
| | 0.75 | 100% |
| | 0.9 | 96% |
| Olympiad | 0.5 | 100% |
| | 0.75 | 95% |
| | 0.9 | 84% |
| Omnimath | 0.5 | 100% |
| | 0.75 | 96 % |
| | 0.9 | 89 % |

in red. The "CauCoT CoT" displays the corrected version produced by CauCoT, where the corrected steps are marked in green.

---

**An Example of the Correction for Measurement Error by CauCoT**

**ID:** GSM8K

**Problem:** A company sold 4000 gallons of milk in jars to Mr. Marcellus' store at the cost of \$3.5 per gallon. However, Mr. Marcellus later realized $\frac{2}{5}$ of the amount of milk he purchased had passed the expiry date and could not be sold. He returned the sour milk to the company and ordered a refund. Calculate how much he got in refunds.

**CRBench CoT (Erroneous Reasoning):**

- **Step 1:** Incorrectly computed expired milk volume as $\frac{2}{5} \times 4000 = 800$ gallons.
- **Step 2:** Refund = $800 \times \$3.5 = \$2800$.
- **Conclusion:** Estimated refund = $\boxed{\$2800}$. ✗**Incorrect due to miscalculation**.

**Error Type:** Measurement Error

**CauCoT CoT (Causal Correction):**

- **Step 1:** Correctly identified expired portion as $\frac{2}{5} \times 4000 = 1600$ gallons.
- **Step 2:** Refund = $1600 \times \$3.5 = \$5600$.
- **Conclusion:** Mr. Marcellus received a refund of $\boxed{\$5600}$ ✓. **Correct**.

---

**An Example of the Correction for Collider Error by CauCoT**

**ID:** Math

**Problem:** Carlos Montado was born on Saturday, November 9, 2002. On what day of the week will Carlos be 706 days old? (Spell out the entire day of the week.)

**CRBench CoT (Erroneous Reasoning):**

- textbfStep 1: Divided $706 \div 7 = 101$ remainder 1, meaning 706 days corresponds to moving **forward** 1 day of the week.

- textbfStep 2: Starting from Saturday and moving 1 day forward gives: Sunday.

- textbfConclusion: Predicted weekday = **Sunday** ✗ **Incorrect**.

**Error Type:** Collider Error

**CauCoT CoT (Causal Correction):**

- **Step 1:** Correct modulo calculation: $706 \div 7 = 100$ remainder 6. Therefore, the weekday should be computed by shifting **forward by 6 days**.
- **Step 2:** Saturday $+ 6$ days = Friday.
- **Conclusion:** Carlos will be 706 days old on a **Friday** ✓ **Correct**.

---

### An Example of the Correction for Mediation Error by CauCoT

**ID:** GSM8K

**Problem:** One dwarf can mine 12 pounds of ore per day with his bare hands. He can mine twice as much with an iron pickaxe and 50% more with a steel pickaxe than with an iron pickaxe. How many pounds of ore can 40 dwarves with steel pickaxes mine in a month with 30 days?

**CRBench CoT (Erroneous Reasoning):**

- **Step 1:** Assumed that steel pickaxes yield 50% more than bare hands: $12 \times 1.5 = 18$ pounds/day.
- **Step 2:** Computed: $18 \times 40 \times 30 = 21,600$ pounds.
- **Conclusion:** Predicted monthly total = **21,600** pounds ✗ **Incorrect**.

**Error Type:** Mediation Error

**CauCoT CoT (Causal Correction):**

- **Step 1:** A dwarf with an iron pickaxe mines $12 \times 2 = 24$ pounds/day.
- **Step 2:** With a steel pickaxe (50% more than iron), a dwarf mines $24 \times 1.5 = 36$ pounds/day.
- **Step 3:** Total for 40 dwarves over 30 days: $36 \times 40 \times 30 = $ **43,200** pounds.
- **Conclusion:** Correct monthly total = **43,200** pounds ✓ **Correct**

---

## D    LLM USAGE

Large Language Models (LLMs) were used to aid in the writing and polishing of the manuscript. Specifically, we used an LLM to assist in refining the language, improving readability, and ensuring clarity in various sections of the paper. The model helped with tasks such as sentence rephrasing, grammar checking, and enhancing the overall flow of the text.

It is important to note that the LLM was not involved in the ideation, research methodology, or experimental design. All research concepts, ideas, and analyses were developed and conducted by the authors. The contributions of the LLM were solely focused on improving the linguistic quality of the paper, with no involvement in the scientific content or data analysis.

The authors take full responsibility for the content of the manuscript, including any text generated or polished by the LLM. We have ensured that the LLM-generated text adheres to ethical guidelines and does not contribute to plagiarism or scientific misconduct.

