# OpenReview forum: "Is It Necessary to Inject Causality into Chain-of-Thought Reasoning?"
_ICLR.cc/2026/Conference — Submitted to ICLR 2026_

### Official Review · Reviewer_5tZu · 2025-10-24

**Soundness:** 3
**Presentation:** 3
**Contribution:** 2
**Rating:** 4
**Confidence:** 4

**Summary:**

This paper explores whether explicit causal modeling is necessary for improving reasoning correctness and interpretability in large language models (LLMs). The authors formalize Chain-of-Thought (CoT) reasoning as a Structural Causal Model (SCM), propose a step-level causal correction algorithm CauCoT, and introduce a causal reasoning benchmark CRBench. The paper claims that explicitly injecting causality into CoT enhances both correctness and interpretability, as evidenced by quantitative improvements in reasoning performance across several models.

**Strengths:**

1. The paper raises an important question about the role of causality in CoT reasoning, which is of clear conceptual and practical interest to the ICLR community.
2. The SCM-based formalization provides a structured framework for analyzing dependencies between reasoning steps.
3. The proposed CauCoT algorithm is well-motivated and could serve as a diagnostic tool for analyzing step-level reasoning errors. The benchmark CRBench introduces a novel evaluation perspective centered on causal error types.

**Weaknesses:**

1. Questionable link between causality and reasoning correctness
    The paper assumes a strong correlation between causal structure and reasoning correctness — that is, explicitly revealing causal dependencies can enhance logical validity. However, this assumption lacks both theoretical justification and empirical evidence. In many reasoning tasks, the so-called 'causal relations' between steps may be merely epiphenomenal rather than truly constitutive: CoT reasoning steps can form a valid sequence through statistical or semantic coherence without reflecting genuine causal dependence. Without formal proofs or ablation studies verifying the correlation between 'causal correctness' and factual or logical correctness, this central claim risks circular reasoning — assuming that correctness derives from causality, while causality itself is defined in terms of correctness.
    The proposed metric CACE relies on two scoring functions, Sans and Slog, which measure causal contribution through textual similarity and logical consistency rather than genuine counterfactual causal inference. As a result, the so-called 'causal effect' likely reflects only surface-level alignment among reasoning steps, rather than true intervention-based causal dependence.

2. Overlap with LLM’s intrinsic mechanisms
    The paper treats 'explicit causal modeling' as an external mechanism added to the reasoning process of LLMs, but overlooks that transformer-based large language models already possess intrinsic causal reasoning capabilities through attention mechanisms and token dependency structures. In Chain-of-Thought (CoT) reasoning, each step is generated conditioned on the previous ones — a process that essentially forms a directed information flow. Although not identical to a Structural Causal Model (SCM), it serves a similar functional role. Therefore, explicitly imposing a causal structure may merely duplicate the dependency patterns that the model has already learned internally.
    Prior research, such as DeCoT, has shown that enhancing causal significance or consistency between reasoning steps can be achieved through representation regularization or consistency loss, without constructing an explicit symbolic causal graph. This paper fails to demonstrate that its SCM-based formalization provides any unique advantage beyond the inherent dependency mechanisms already present in LLMs. To substantiate the central claim that 'injecting causality is necessary', the authors should show that:
    (a) the LLM’s implicit dependency graph substantially differs from the SCM-derived causal graph; and
    (b) enforcing the latter effectively closes a measurable reasoning performance gap.
    However, the paper provides neither analysis. Without such comparison, the contribution appears to be more of a causal rephrasing of existing CoT dependencies rather than a genuinely new mechanism.

3. Coverage of experimental results
   The authors claim that CauCoT improves reasoning correctness and interpretability across various LLMs; however, the experiments are conducted only on open-source small to mid-sized models such as LLaMA and Qwen, without including the latest high-performance models like GPT-4/4o, Gemini-2.5, or Claude-3. Therefore, the generality of the proposed algorithm for stronger models remains unproven. It is recommended that the authors include comparative experiments or qualitative analyses on these frontier models.


4. Unclear Expression
    The description of Table 2 is inconsistent: the text states that Columns 2–8 are zero-shot baselines and that the last two columns correspond to 'fine-tuned' CauCoT results, yet the authors also mention highlighting 'the top three zero-shot results per block,' where CauCoT entries are color-marked as well.

**Questions:**

1. Can you provide empirical evidence that causal correctness (as defined by CACE metric) positively correlates with logical or factual correctness across tasks?
2. How to distinguish between causal dependencies explicitly modeled in SCM and the conditional dependencies already inherent in autoregressive LLM reasoning?
3. The authors may refer to the following papers: https://arxiv.org/abs/2402.16048v1,  https://aclanthology.org/2024.sighan-1.17, https://aclanthology.org/2024.acl-long.758

---

### Official Review · Reviewer_dYUW · 2025-10-26

**Soundness:** 3
**Presentation:** 3
**Contribution:** 3
**Rating:** 4
**Confidence:** 4

**Summary:**

In this paper, the authors formailize the chain of thought (CoT) process as a structural causal model, representing the reasoning process as a causal graph. The authors propose a new algorithm CauCoT which aims to correct erroneous steps in CoT until all steps are causally correct. To better evaluate the performance, the authors also present a new benchmark dataset called CRBench which consists of both real and generated scenarios.

**Strengths:**

1. CoT research is becoming increasingly popular and relevant these days and is an appropriate topic for the venue.
2. The contribution of the new CRBench benchmark enables further research
3. The figures explaining the model/architecture are very helpful to the reader

**Weaknesses:**

1. Some of the important parts that are crucial for understanding (the faith metric, for example) the work better have been relegated to the appendix. I understand not everything can be fit into the main section but some effort to include a short summary/essence of the topic would be appreciated.

2. It is not clear how the labeling and generation of the benchmark dataset is done. Is this manually done by humans or using LLMs? If the latter, wouldn't using LLMs to influence causal data affect how CauCoT's (or any other future method's) performance is measured?

3. Have the authors considered any prompt/response variance in LLMs that can affect the metrics?

**Questions:**

1. Any reason why CauCoT performs much worse with Llama as shown in Table 2?
2. Some ablation studies with CauCoT would shed light on which components influence the performance the most

---

### Official Review · Reviewer_jSrm · 2025-10-30

**Soundness:** 2
**Presentation:** 2
**Contribution:** 3
**Rating:** 2
**Confidence:** 4

**Summary:**

This paper proposes CauCoT, a method to identify causally erroneous steps in Chain-of-Thought (CoT) and iteratively updates them until all steps are causally correct. To evaluate the effectiveness of CauCoT, the paper also creates CRBench, a benchmark for identifying and correcting causal errors in CoT. Experiments on CRBench show that causally informed reasoning improves both correctness and interpretability of CoT reasoning.

**Strengths:**

1.	The paper proposes a framework for analyzing CoT reasoning errors using structural causal models, along with a method to identify and correct these errors.
2.	The creation of CRBench as a benchmark for diagnosing causal reasoning errors in CoT reasoning provides a useful tool for future research on evaluating reasoning failures.

**Weaknesses:**

1. The paper does not discuss related work on analyzing and improving CoT reasoning through causal inference, such as [1] and [2]. A dedicated related work section should be added to introduce and contextualize relevant literature.

[1] Paul, Debjit, et al. "Making Reasoning Matter: Measuring and Improving Faithfulness of Chain-of-Thought Reasoning." Findings of the Association for Computational Linguistics: EMNLP 2024. 2024.

[2] Bao, Guangsheng, et al. "How Likely Do LLMs with CoT Mimic Human Reasoning?." Proceedings of the 31st International Conference on Computational Linguistics. 2025.

2. The definition of the CoT Average Causal Effect (ATE) is unclear. The “logical effect,” which measures the incremental coherence of ci attributable to its parent nodes, appears to describe the contribution of ci’s parents rather than the ATE of ci itself.
3. The experimental setup could be improved. CauCoT is applied to CRBench to produce causally corrected traces, and these are used to fine-tune open-source LLMs; however, the baseline methods are based on non-fine-tuned LLMs. As a result, it is difficult to assess CauCoT’s effectiveness from the reported results. It would be more informative to (1) separately evaluate CauCoT’s ability to identify and correct causal reasoning errors, and (2) compare the fine-tuned models’ performance against other fine-tuned baselines.
4. The presentation can be improved:

(a) Citations are not formatted correctly.

(b) Details of the fine-tuning procedure, such as learning rate and number of epochs, are missing.

(c) Figure 2: Conclution -> Conclusion

(d) Figure 3 caption: a CoT achieve -> a CoT achieves

(e) It is unconventional to use do(ci) to represent removing the influence of ci and do(∅) to indicate no intervention. Typically, the do-calculus is not used when no intervention is performed.

**Questions:**

1.	What is the difference between Definition 2 and the last sentence of Definition 1?
2.	Does the paper demonstrate that incorporating causality into CoT reasoning is necessary, or simply that it is beneficial?

---

### Official Review · Reviewer_YjY1 · 2025-11-01

**Soundness:** 2
**Presentation:** 1
**Contribution:** 1
**Rating:** 2
**Confidence:** 3

**Summary:**

The paper investigates the question of whether explicitly modeling causality in CoT reasoning for LLMs is beneficial. The authors propose a formalization of CoT as a Structural Causal Model (SCM), where each reasoning step is treated as a causal variable with parent links corresponding to prior steps or external knowledge. Based on this formalization, they define a metric, the “Chain-of-Thought Average Causal Effect” (CACE), which quantifies the evidential contribution and logical coherence of each reasoning step. Using this, they develop an algorithm (CauCoT) that iteratively identifies "causally weak" steps, generates revised candidate steps via LLM prompting, selects improvements, and repeats until the chain satisfies a threshold of “causal correctness”. They also introduce a benchmark (CRBench) of reasoning traces annotated w.r.t. causal‐error types (e.g., measurement error, collider bias, confounding, mediation error). They empirically evaluate on a set of reasoning tasks (mathematical, commonsense) and show that applying their method improves accuracy and a “faithfulness” interpretability metric relative to standard CoT or other baselines. The main claim is that injecting causality into CoT is necessary (or at least strongly beneficial) for more correct, interpretable reasoning in LLMs.

**Strengths:**

1. Well-structured writing.

2. Provides explicit formulas for causal scoring (γₑ, γₗ, γ_{CoT}).

3. A large synthetic benchmark (CRBench ~12k samples).

4. tackles interpretability, not just accuracy.

**Weaknesses:**

1. Causal semantics are largely nominal, not identified:
CACE relies on textual ablations/re-generations scored by task-specific Sans/Slog; these are not identifiable counterfactuals and inherit model/heuristic bias. The paper acknowledges textual “do” approximations and their limits (Appendix C.1), undercutting the central causal claim. In short, γ_{CoT} measures perturbation sensitivity, not causal effect in the Pearl/Rubin sense.

2. Benchmark realism is weak; clear overfitting risk:
CRBench is largely synthetically injected errors guided by four templates. which makes the evaluation tightly coupled to the method’s own error taxonomy rather than to naturally occurring LLM failures. No evidence is provided that these four classes dominate real error modes, yet all generated splits are 100% wrong by construction.

3. Claims outpace evidence, strongest model regresses:
The paper claims causality is “necessary,” but ToT outperforms CauCoT on Qwen2.5-72B-Inst (50.5 vs 47.6 Acc), which is exactly the regime practitioners care about. This contradicts the necessity framing and suggests benefits do not scale with capability.

4. Limited novelty vs. established critique-and-revise paradigms:
CauCoT’s loop (judge -> rewrite -> select) resembles existing self-critique/self-refinement methods (e.g. 1,2), while ToT already performs structured step exploration. What is new is primarily the terminology (SCM/CACE), not the control structure. The paper does not position itself against these baselines beyond ToT/SC-CoT.

(1) https://arxiv.org/abs/2303.11366

and e.g. (2) https://anote-ai.medium.com/recursive-criticism-and-improvlanguage-models-can-solve-computer-tasks-7285b4cea339



5. Faithfulness/interpretability remain model-internal:
"Faith" is a model-driven score with no human study or external ground truth; recent work shows CoT faithfulness is fragile, and causal-mediation analyses find models often don’t use their written steps. The paper does not reconcile with this literature.

6. Compute/latency and sensitivity are underspecified:
CauCoT requires multiple Monte-Carlo rollouts and iterative rewrites per step, but there’s no wall-clock, FLOPs, or budget-vs-quality ablation, nor a σ/α/β sensitivity summary in the main text. This obscures practical feasibility.

**Questions:**

1. What makes your "causal effect" truly causal and not just a sensitivity score?
   - How do you justify calling γ_CoT an intervention-based effect if no counterfactual semantic model of the LLM is used?

2. How realistic are your injected error types in CRBench?
   - Can you show statistics from naturally occurring chain-of-thought errors produced by GPT/Qwen/PaLM that match “measurement/collider/confounding/mediation” classes?

3. Why does CauCoT harm performance on the strongest model (Qwen2.5-72B)?
   - Is the issue overcorrection, hallucination during rewriting, or error accumulation? What failure analysis did you perform?

4. How does your method differ mechanistically from Reflexion (1) or RCI prompting (e.g. 2)?
   - They also detect faulty intermediate steps and rewrite reasoning. What exactly is new besides causal terminology?

5. Did you test generalization outside CRBench?
   - For example: GSM8K raw CoT traces, StrategyQA, Minerva (3, a model that can evaluate these kinds of CoT), or human-written CoT. Or does your method only improve synthetic error-injected traces?

(1) https://arxiv.org/abs/2303.11366
(2) https://anote-ai.medium.com/recursive-criticism-and-improvlanguage-models-can-solve-computer-tasks-7285b4cea339
(3) https://research.google/blog/minerva-solving-quantitative-reasoning-problems-with-language-models/?utm_source=chatgpt.com

6. Do human annotators agree that edited reasoning chains are more coherent and causally valid, or is it only a model saying so?

---

### Meta-Review · Area_Chair_JPHf · 2026-01-03

**Summary:**

This paper explores whether explicitly modeling causality can improve Chain-of-Thought reasoning in LLMs. To this end, the authors formalize the CoT as a structural causal model and  develop a step-level causal correction algorithm, Causalizing Chain-of-Thought (CauCoT). In addition, the work release the Causal Reasoning Benchmark (CRBench) for identifying and correcting causal errors in CoT.

However, the reviewers raised concerns about the paper’s motivation and its comparison with related work, among other issues. The authors did not provide any rebuttal.

**Reviewer Concerns:**

No rebuttal

**Reviewer Scores:**

No rebuttal

---

### Decision · Program_Chairs · 2026-01-26

Reject